# TOWARDS UNDERSTANDING GD WITH HARD AND CONJUGATE PSEUDO-LABELS FOR TEST-TIME ADAPTATION

**Jun-Kun Wang and Andre Wibisono**
Department of Computer Science, Yale University
{jun-kun.wang,andre.wibisono}@yale.edu

## ABSTRACT

We consider a setting that a model needs to adapt to a new domain under distribution shifts, given that only unlabeled test samples from the new domain are accessible at test time. A common idea in most of the related works is constructing pseudo-labels for the unlabeled test samples and applying gradient descent (GD) to a loss function with the pseudo-labels. Recently, Goyal et al. (2022) propose conjugate labels, which is a new kind of pseudo-labels for self-training at test time. They empirically show that the conjugate label outperforms other ways of pseudo-labeling on many domain adaptation benchmarks. However, provably showing that GD with conjugate labels learns a good classifier for test-time adaptation remains open. In this work, we aim at theoretically understanding GD with hard and conjugate labels for a binary classification problem. We show that for square loss, GD with conjugate labels converges to an $\epsilon$-optimal predictor under a Gaussian model for any arbitrarily small $\epsilon$, while GD with hard pseudo-labels fails in this task. We also analyze them under different loss functions for the update. Our results shed lights on understanding when and why GD with hard labels or conjugate labels works in test-time adaptation.

## 1 INTRODUCTION

Fully test-time adaptation is the task of adapting a model from a source domain so that it fits to a new domain at test time, without accessing the true labels of samples from the new domain nor the data from the source domain (Goyal et al., 2022; Wang et al., 2021a; Li et al., 2020; Rusak et al., 2021; Zhang et al., 2021a; S & Fleuret, 2021; Mummadi et al., 2021; Iwasawa & Matsuo, 2021; Liang et al., 2020; Niu et al., 2022; Thopalli et al., 2022; Wang et al., 2022b; Kurmi et al., 2021). Its setting is different from many works in domain adaptation or test-time training, where the source data or statistics of the source data are available, e.g., Xie et al. (2021); Liu et al. (2021a); Prabhu et al. (2021); Sun et al. (2020); Chen et al. (2022); Hoffman et al. (2018); Eastwood et al. (2022); Kundu et al. (2020); Liu et al. (2021b); Schneider et al. (2020); Gandelsman et al. (2022); Zhang et al. (2021b); Morerio et al. (2020); Su et al. (2022). Test-time adaptation has drawn growing interest recently, thanks to its potential in real-world applications where annotating test data from a new domain is costly and distribution shifts arise at test time due to some natural factors, e.g., sensor degradation (Wang et al., 2021a), evolving road conditions (Gong et al., 2022; Kumar et al., 2020), weather conditions (Bobu et al., 2018), or change in demographics, users, and time periods (Koh et al., 2021).

The central idea in many related works is the construction of the pseudo-labels or the proposal of the self-training loss functions for the unlabeled samples, see e.g., Wang et al. (2021a); Goyal et al. (2022). More precisely, at each test time $t$, one receives some unlabeled samples from a new domain, and then one constructs some pseudo-labels and applies a GD step to the corresponding self-training loss function, as summarized in Algorithm 1. Recently, Goyal et al. (2022) propose a new type of pseudo-labels called conjugate labels, which is based on an observation that certain loss functions can be naturally connected to conjugate functions, and the pseudo-labels are obtained by exploiting a property of conjugate functions (to be elaborated soon). They provide a modular approach of constructing conjugate labels for some loss functions, e.g., square loss, cross-entropy loss, exponential loss. An interesting finding of Goyal et al. (2022) is that a recently proposed self-training loss for test-time adaptation of Wang et al. (2021a) can be recovered from their conjugate-label

---

**Algorithm 1:** Test-time adaptation via pseudo-labeling

---

1: **Init:** $w_1 = w_\mathcal{S}$, where $w_\mathcal{S}$ is the model learned from a source domain.
2: **Given:** Access to samples from the data distribution $D_{\text{test}}$ of a new domain.
3: **for** $t = 1, 2, \ldots, T$ **do**
4:     Get a sample $x_t \sim D_{\text{test}}$ from the new domain.
5:     Construct a pseudo-label $y_{w_t}^{\text{pseudo}}(x_t)$ and consequently a self-training loss function $\ell^{\text{self}}(w_t; x_t)$.
6:     Apply gradient descent (GD): $w_{t+1} = w_t - \eta \nabla_w \ell^{\text{self}}(w_t; x_t)$.
7: **end for**

---

framework. They also show that GD with conjugate labels empirically outperforms that of other pseudo-labels like hard labels and robust pseudo-labels (Rusak et al., 2021) across many benchmarks, e.g., ImageNet-C (Hendrycks & Dietterich, 2019), ImageNet-R (Hendrycks et al., 2021), VISDA-C (Peng et al., 2017), MNISTM (Ganin & Lempitsky, 2015). However, certain questions are left open in their work. For example, why does GD with conjugate labels work? Why can it dominate GD with other pseudo-labels? To our knowledge, while pseudo-labels are quite indispensable for self-training in the literature (Li et al., 2019; Zou et al., 2019), works that theoretically understand the dynamic of GD with pseudo-labels are very sparse, and the only work that we are aware is of Chen et al. (2020). Chen et al. (2020) show that when data have spurious features, if *projected* GD is initialized with sufficiently high accuracy in a new domain, then by minimizing the exponential loss with hard labels, projected GD converges to an approximately Bayes-optimal solution under certain conditions. In this work, we study vanilla GD (without projection) for minimizing the self-training loss derived from square loss, logistic loss, and exponential loss under hard labels and conjugate labels.

We prove a performance gap between GD with conjugate labels and GD with hard labels under a simple Gaussian model (Schmidt et al., 2018; Carmon et al., 2019). Specifically, we show that GD with hard labels for minimizing square loss can not converge to an $\epsilon$-optimal predictor (see (8) for the definition) for any arbitrarily small $\epsilon$, while GD with conjugate labels converge to an $\epsilon$-optimal predictor exponentially fast. Our theoretical result champions the work of conjugate labels of Goyal et al. (2022). We then analyze GD with hard and conjugate labels under logistic loss and exponential loss, and we show that under these scenarios, they converge to an optimal solution at a $\log(t)$ rate, where $t$ is the number of test-time iterations. Our results suggest that the performance of GD in test-time adaptation depends crucially on the choice of pseudo-labels and loss functions. Interestingly, the problems of minimizing the associated self-training losses of conjugate labels in this work are non-convex optimization problems. Hence, our theoretical results find an application in non-convex optimization where GD can enjoy some provable guarantees.

## 2 PRELIMINARIES

We now give an overview of hard labels and conjugate labels. But we note that there are other proposals of pseudo-labels in the literature. We refer the reader to Li et al. (2019); Zou et al. (2019); Rusak et al. (2021) and the references therein for details.

**Hard labels:** Suppose that a model $w$ outputs $h_w(x) \in \mathbb{R}^K$ and that each element of $h_w(x)$ could be viewed as the predicted score of each class for a multi-class classification problem with $K$ classes. A hard pseudo-label $y_w^{\text{hard}}(x)$ is a one-hot vector which is 1 on dimension $k$ (and 0 elsewhere) if $k = \arg\max_k h_w(x)[k]$, i.e., class $k$ has the largest predicted score by the model $w$ for a sample $x$ (Goyal et al., 2022). On the other hand, for a binary classification problem by a linear predictor, i.e., $h_w(x) = w^\top x$, a hard pseudo-label is simply defined as:

$$y_w^{\text{hard}}(x) := \text{sign}(w^\top x), \tag{1}$$

see, e.g., Kumar et al. (2020), Chen et al. (2020). GD with hard labels is the case when Algorithm 1 uses a hard label to construct a gradient $\nabla_w \ell^{\text{self}}(w_t; x_t)$ and update the model $w$.

**Conjugate labels (Goyal et al., 2022):** The approach of using conjugate labels as pseudo-labels crucially relies on the assumption that the original loss function is of the following form:

$$\ell(w; (y, x)) := f(h_w(x)) - y^\top h_w(x), \tag{2}$$

where $f(\cdot) : \mathbb{R}^K \to \mathbb{R}$ is a scalar-value function, and $y \in \mathbb{R}^K$ is the label of $x$, which could be a one-hot encoding vector in multi-class classification. Since the true label $y$ of a sample $x$ is not

Table 1: Summary of {Hard, Conjugate} pseudo-labels and the resulting self-training loss functions using square loss, logistic loss, and exponential loss.

| **Square loss:** $\ell^{\mathrm{exp}}(w;(x,y)) := \frac{1}{2}(y - w^\top x)^2$. | | |
|---|---|---|
| Hard | $y_w^{\mathrm{hard}}(x) = \mathrm{sign}(w^\top x)$ | $\ell^{\mathrm{hard}}(w;x) = \frac{1}{2}(\mathrm{sign}(w^\top x) - w^\top x)^2$ |
| Conjugate | $y_w^{\mathrm{conj}}(x) = w^\top x$ | $\ell^{\mathrm{conj}}(w;x) = -\frac{1}{2}(w^\top x)^2$ |
| **Logistic loss:** $\ell^{\mathrm{logit}}(w;(x,y)) := \log\left(\cosh\left(w^\top x\right)\right) - y(w^\top x)$, where $y = \{+1, -1\}$. | | |
| Hard | $y_w^{\mathrm{hard}}(x) = \mathrm{sign}(w^\top x)$ | $\ell^{\mathrm{hard}}(w;x) = \log\left(\cosh\left(w^\top x\right)\right) - |w^\top x|$ |
| Conjugate | $y_w^{\mathrm{conj}}(x) = \tanh\left(w^\top x\right)$ | $\ell^{\mathrm{conj}}(w;x) = \log\left(\cosh\left(w^\top x\right)\right) - \tanh\left(w^\top x\right)w^\top x$ |
| **Exponential loss:** $\ell^{\mathrm{exp}}(w;(x,y)) := \exp(-yw^\top x)$, where $y = \{+1, -1\}$. | | |
| Hard | $y_w^{\mathrm{hard}}(x) = \mathrm{sign}(w^\top x)$ | $\ell^{\mathrm{hard}}(w;x) = \exp(-|w^\top x|)$ |
| Conjugate | $y_w^{\mathrm{conj}}(x) = \tanh\left(w^\top x\right)$ | $\ell^{\mathrm{conj}}(w;x) = \mathrm{sech}\left(w^\top x\right)$ |

available in test-time adaptation, it is natural to construct a pseudo-label $y_w^{\mathrm{pseudo}}(x)$ and consequently a self-training loss function by replacing $y$ with $y_w^{\mathrm{pseudo}}(x)$ in (2),

$$\ell^{\mathrm{conj}}(w;x) := f(h_w(x)) - y_w^{\mathrm{pseudo}}(x)^\top h_w(x). \tag{3}$$

One can then compute the gradient $\nabla\ell^{\mathrm{conj}}(w;(y,x))$ and use GD to adapt the model $w$ at test time.

Define $h_* \in \mathbb{R}^K$ as $h_* \leftarrow \arg\min_{h \in \mathbb{R}^K} f(h) - y^\top h$, where $-f^*(y) = \min_{h \in \mathbb{R}^K} f(h) - y^\top h$ is the conjugate function, see e.g, Chapter 3.3 in Boyd et al. (2004). It turns out that $h_*$ satisfies $y = \nabla f(h_*)$. From the similarity, Goyal et al. (2022) propose conjugate labels:

$$y_w^{\mathrm{conj}}(x) := \nabla f(h_w(x)), \tag{4}$$

where $y_w^{\mathrm{conj}}(x)$ is possibly a real-value vector instead of a one-hot encoding vector. Let $y_w^{\mathrm{pseudo}}(x) \leftarrow y_w^{\mathrm{conj}}(x)$ in (3). Then, we get the self-training loss function using the conjugate label:

$$\ell^{\mathrm{conj}}(w;x) := f(h_w(x)) - \nabla f(h_w(x))^\top h_w(x). \tag{5}$$

We note that GD with conjugate labels is an instance of Algorithm 1 when we let $\nabla_w \ell^{\mathrm{self}}(w_t;x_t) \leftarrow \nabla_w \ell^{\mathrm{conj}}(w_t;x_t)$ at each test time $t$.

Table 1 summarizes conjugate labels and hard labels as well as their self-training loss functions using square loss, logistic loss, and exponential loss. We provide the derivation of the case using square loss below, while the rest of them are available in Appendix A.
**(Square loss) Example of a conjugate label $y_w^{\mathrm{conj}}(x)$ and its self-training function $\ell^{\mathrm{conj}}(w;x)$:** Observe that square loss $\ell(w;(x,y)) := \frac{1}{2}(y - w^\top x)^2$ is in the form of (2) up to a constant, where $f(\cdot) = \frac{1}{2}(\cdot)^2 : \mathbb{R} \to \mathbb{R}^+$. Substituting $f(\cdot) = \frac{1}{2}(\cdot)^2$ and $h(w) = w^\top x$ in (4) and (5), we get

$$y_w^{\mathrm{conj}}(x) = w^\top x, \quad \text{and} \quad \ell^{\mathrm{conj}}(w;x) = -\frac{1}{2}(w^\top x)^2. \tag{6}$$

## 3   THEORETICAL FRAMEWORK: GAUSSIAN MODEL

Our theoretical analysis considers a binary classification setting in which samples from the new domain are generated as $x \sim \mathcal{N}(y\mu, \sigma^2 I_d) \in \mathbb{R}^d$, where $\mu \in \mathbb{R}^d$ is the mean and $\sigma^2 > 0$ is the magnitude of the covariance. The label $y$ is assumed to be uniform on $\{-1, 1\}$. Therefore, we have $P(X|Y = y) = \mathcal{N}(y\mu, \sigma I_d)$ and $P(y = -1) = P(y = 1) = \frac{1}{2}$ under Gaussian model (Schmidt et al., 2018; Carmon et al., 2019; Kumar et al., 2020).

Given a test sample $x$, a linear predictor $w \in \mathbb{R}^d$ makes a prediction of the label $\hat{y}_w(x)$ as $\hat{y}_w(x) = \mathrm{sign}(w^\top x)$. While a model could be self-trained under various loss functions, the natural metric to evaluate a model for classification is the expected 0-1 loss. Under Gaussian model, the expected 0-1 loss enjoys a simple closed-form expression:

$$\ell^{0-1}(w) := \mathbb{E}_{(x,y)}[\mathbb{1}\{y\hat{y}_w(x) \neq 0\}] = P[yw^\top x < 0] = P\left(N\left(\frac{\mu^\top w}{\sigma\|w\|}, 1\right) < 0\right) = \Phi\left(\frac{\mu^\top w}{\sigma\|w\|}\right), \tag{7}$$

where $\Phi(u) := \frac{1}{\sqrt{2\pi}} \int_u^\infty \exp(-z^2/2)dz$ is the Gaussian error function. From (7), one can see that the predictors that minimize the $0-1$ loss are those that align with $\mu$ in direction and the minimum error is $\Phi\left(\frac{\|\mu\|}{\sigma}\right)$. In other words, an optimal linear predictors $w_* \in \mathbb{R}^d$ has to satisfy $\cos\left(\frac{w_*}{\|w_*\|}, \frac{\mu}{\|\mu\|}\right) = 1$.

In our theoretical analysis, we let $\mu = [\|\mu\|, 0, \ldots, 0]^\top \in \mathbb{R}^d$; namely, the first element is the only non-zero entry. Our treatment is without loss of generality, since we can rotate and change a coordinate system if necessary. For any vector $w \in \mathbb{R}^d$, its orthogonal component to $\mu$ is $\left(I_d - \frac{\mu}{|\mu|}\frac{\mu^\top}{|\mu|}\right)w$. Thanks to the assumption of $\mu$, the orthogonal space (to $\mu$) is the subspace of dimension 2 to $d$. Indeed, for any vector $w$, its orthogonal component (to $\mu$) $\left(I_d - \frac{\mu}{|\mu|}\frac{\mu^\top}{|\mu|}\right)w$ is always $0$ in its first entry. Therefore, we can represent an orthogonal component of $w$ as $[w[2], \ldots, w[d]] \in \mathbb{R}^{d-1}$.

> We call a model $w \in \mathbb{R}^d$ an $\epsilon$-**optimal predictor** under Gaussian model if it satisfies two conditions:
>
> **Condition 1:** $\quad \left\langle w, \frac{\mu}{\|\mu\|} \right\rangle = w[1] > 0 \quad$ and $\quad$ **Condition 2:** $\quad \cos^2\left(\frac{w}{\|w\|}, \frac{\mu}{\|\mu\|}\right) \geq 1 - \epsilon.$
>
> $\hspace{12cm}$ (8)

Using (7), the expected $0-1$ loss of an $\epsilon$-optimal predictor is $\ell^{0-1}(w) = \Phi\left(\frac{\|\mu\|}{\sigma}\sqrt{1-\epsilon}\right)$. To get an $\epsilon$-optimal predictor, we need to satisfy $\langle w, \mu \rangle > 0$ and also need that the ratio of the projection onto $\mu$ to the size of the orthogonal component to $\mu$ is as large as possible, i.e., $\frac{w[1]^2}{\sum_{i\neq 1}^d w^2[i]}$ is large, which can be seen from the following equalities: $\cos^2\left(\frac{w}{\|w\|}, \frac{\mu}{\|\mu\|}\right) = \frac{\langle w,\mu \rangle^2}{\|w\|^2\|\mu\|^2} = \frac{w[1]^2}{\sum_{i=1}^d w[i]^2} = \frac{1}{1+\frac{\sum_{i\neq 1}^d w[i]^2}{w[1]^2}}.$

The projection of $w$ onto $\mu$ has to be positive and large when the size of the orthogonal component is non-zero to get an $\epsilon$-optimal predictor, i.e., $w[1] \gg 0$.

Finally, in our analysis we will assume that the initial point satisfies Condition 1 on (8), which means that the initial point forms an acute angle with $\mu$. This is a mild assumption, as it means that the source model is better than the random guessing in the new domain.

**Related works of Gaussian model:** In recent years, there are some works that adopt the framework of Gaussian model to show some provable guarantees under various topics. For example, Schmidt et al. (2018) and Carmon et al. (2019) studying it for adversarial robustness. For another example, Kumar et al. (2020) recently show that self-training with hard labels can learn a good classifier when infinite unlabeled data are available and that the distributions shifts are mild. Their theoretical result perhaps is the most relevant one to ours in the literature, in addition to Chen et al. (2020) that we have discussed in the introduction. Kumar et al. (2020) consider the setting of gradual distribution shifts so that the data distribution in each iteration $t$ is different and that the update in each $t$ is a minimizer of a constrained optimization:

$$w_t \leftarrow \arg\min_{w\in\Theta} \mathbb{E}_{x\sim D_t}\left[L\left(y_w^{\text{hard}}(x)w^\top x\right)\right], \text{ where } \Theta := \left\{w : \|w\| \leq 1, \|w - w_{t-1}\| \leq \tfrac{1}{2}\right\}.$$
$$\hspace{14cm}(9)$$

On (9), $L(\cdot) : \mathbb{R} \to \mathbb{R}^+$ is a continuous decreasing function, $D_t$ represents the data distribution at $t$, and $y_w^{\text{hard}}(x) := \text{sign}(w^\top x)$ is the hard label for an unlabeled sample $x$. The main message of their result is that even though the data distribution of the target domain could be very different from that of the source domain, by using data from the intermediate distributions that change gradually, a good classifier for the target domain can be obtained in the end. On the other hand, we consider analyzing GD with pseudo-labels at test-time iterations, and we do not assume that there are intermediate distributions. Our goal is to provably show that GD with pseudo-labels can learn an optimal classifier in a new domain when only unlabeled samples are available at test time, which is different from the setup of Kumar et al. (2020) that simply assumes the access to a minimizer of a certain objective.

## 4 (A NEGATIVE EXAMPLE) GD WITH HARD LABELS UNDER SQUARE LOSS

One of the common loss function is square loss. Recent works have shown that even for the task of classification, a model trained under square loss can achieve competitive performance for classification

as compared to that of a model trained under certain classification losses like cross-entropy loss (Demirkaya et al., 2020; Han et al., 2022; Hui & Belkin, 2020). In this section, we analyze test-time adaptation by GD with hard pseudo-labels under square loss. Recall the definition of square loss: $\ell(w; (x, y)) = \frac{1}{2}(y - w^\top x)^2$. By using hard labels as (1), the self-training loss function becomes

$$\ell^{\mathrm{hard}}(w; x) := \frac{1}{2}\left(y_w^{\mathrm{hard}}(x) - w^\top x\right)^2 = \frac{1}{2}\left(\mathrm{sign}(w^\top x) - w^\top x\right)^2. \tag{10}$$

It is noted that the derivative of $\mathrm{sign}(\cdot)$ is 0 everywhere except at the origin. Furthermore, $\mathrm{sign}(\cdot)$ is not differentiable at the origin. Define $\mathrm{sign}(0) = 0$. Then, $\mathrm{sign}(w^\top x) - w^\top x = 0$ when $w^\top x = 0$, which allows us to avoid the issue of the non-differentiability. Specifically, we can write the gradient as $\nabla \ell^{\mathrm{hard}}(w; x) = -\left(\mathrm{sign}(w^\top x) - w^\top x\right) x$. Using the gradient expression, we obtain the dynamic of GD with hard labels under square loss,

$$w_{t+1} = w_t - \eta \nabla \ell^{\mathrm{hard}}(w_t; x_t) = w_t + \eta \left(\mathrm{sign}(w_t^\top x_t) - w_t^\top x_t\right) x_t. \tag{11}$$

What we show in the following proposition is that the update $w_t$ of (11) does not converge to the class mean $\mu$ in direction. However, it should be noted that a perfect classifier (i.e., one that has the zero 0-1 loss) does not necessarily need to align with the class mean $\mu$ depending on the setup.

**Proposition 1.** *GD with hard labels using square loss fails to converge to an $\epsilon$-optimal predictor for any arbitrarily small $\epsilon > 0$ even under the noiseless setting of Gaussian model ($\sigma = 0$). More precisely, we have $\cos\left(\frac{w_t}{\|w_t\|}, \frac{\mu}{\|\mu\|}\right) \leq 1 - \bar{\epsilon}$, for some $\bar{\epsilon} > 0$ as $t \to \infty$ if $w_\infty$ exists.*

*Proof.* In this proof, we denote $\bar{a}_t := w_t[1] = \left\langle w_t, \frac{\mu}{\|\mu\|} \right\rangle$. From (11), we have

$$\bar{a}_{t+1} = \bar{a}_t + \eta \left(\mathrm{sign}(w_t^\top x_t) - w_t^\top x_t\right) \left\langle x_t, \frac{\mu}{\|\mu\|} \right\rangle. \tag{12}$$

Let us consider the simple noiseless setting of Gaussian model, i.e., $\sigma = 0$, as we aim at giving a non-convergence example. Then, we have $x_t = y_t \mu$ and the dynamic (12) becomes

$$\bar{a}_{t+1} = (1 - \eta \|\mu\|^2)\bar{a}_t + \eta \, \mathrm{sign}(\bar{a}_t \|\mu\|) \|\mu\|, \tag{13}$$

where we used $y_t^2 = 1$ and $y_t \, \mathrm{sign}(y_t \cdot) = \mathrm{sign}(\cdot)$ because $y_t = \{-1, +1\}$.

**Case:** $\eta \leq \frac{1}{\|\mu\|^2}$: Given the initial condition $\bar{a}_1 > 0$, we have $\bar{a}_t > 0, \forall t$ from (13), and $\mathrm{sign}(\bar{a}_t \|\mu\|) = 1, \forall t$. Then, we can recursively expand (13) from time $t + 1$ back to time 1 and obtain

$$\bar{a}_{t+1} = (1 - \eta \|\mu\|^2)^t \bar{a}_1 + \eta \|\mu\| \sum_{s=0}^t (1 - \eta \|\mu\|^2)^s. \tag{14}$$

From (14), we know that $\bar{a}_t \to \frac{1}{\|\mu\|}$, as $t \to \infty$, where we used that $\sum_{s=0}^\infty (1 - \eta \|\mu\|^2)^s = \frac{1}{\eta \|\mu\|^2}$. On the other hand, the dynamic of the orthogonal component $i \neq 1 \in [d]$ is

$$w_{t+1}[i] = w_t[i] + \eta \left(\mathrm{sign}(w_t^\top x_t) - w^\top x_t\right) x[i] = w_t[i], \tag{15}$$

where in the last equality we used that $x_t = y_t \mu$ and $\mu = [\|\mu\|, 0, \ldots, 0]^\top \in \mathbb{R}^d$ so that $x[i] = 0, \forall i \neq 1$. By (14) and (15), we get $\frac{\sum_{i \neq 1}^d w_\infty[i]^2}{w_\infty[1]^2} = \frac{\sum_{i \neq 1}^d w_1[i]^2}{1/\|\mu\|^2}$. That is, the ratio converges to a non-zero value, which implies that GD with hard labels fails to converge to an $\epsilon$-optimal predictor for any arbitrarily small $\epsilon$, i.e., $\cos\left(\frac{w_\infty}{\|w_\infty\|}, \frac{\mu}{\|\mu\|}\right) \leq 1 - \bar{\epsilon}$ for some $\bar{\epsilon} > 0$.

**Case:** $\eta > \frac{1}{\|\mu\|^2}$: Suppose $\bar{a}_t > 0$. Then, the condition that $\bar{a}_{t+1} \geq \bar{a}_t$ is $\frac{1}{\|\mu\|} \geq \bar{a}_t$ from (13), which means that the projection to $\mu$ is bounded and hence the model $w_t$ cannot be an $\epsilon$-optimal classifier for any arbitrarily small $\epsilon$. On the other hand, if $\bar{a}_t > \frac{1}{\|\mu\|}$, then $\bar{a}_{t+1} < \bar{a}_t$, and $\bar{a}_{t+1}$ could even be negative when $\bar{a}_t > \frac{1}{\|\mu\| - 1/(\eta\|\mu\|)}$. Moreover, if $\eta > \frac{2}{\|\mu\|^2}$ and $|\bar{a}_t| > \frac{\eta\|\mu\|}{\eta\|\mu\|^2 - 2} = \frac{1}{\|\mu\| - 2/(\eta\|\mu\|)}$, then the magnitude $|\bar{a}_t|$ is increasing and the sign of $\bar{a}_t$ is oscillating; more precisely, we will have $|\bar{a}_{t+1}| \geq |\bar{a}_t|$ and $\mathrm{sign}(\bar{a}_{t+1}) = -\mathrm{sign}(\bar{a}_t)$. Consequently, the model $w_t$ is not better than the random guessing at every other iteration (recall (7)), which is not desirable for test-time adaptation. □

In the next section, we will provably show that GD with conjugate labels under square loss can learn an $\epsilon$-optimal predictor for any arbitrary $\epsilon$, which is the first theoretical result in the literature that shows the advantage of conjugate labels over hard labels, to the best of our knowledge.

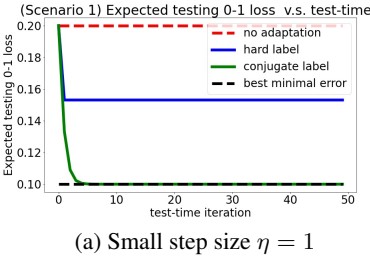 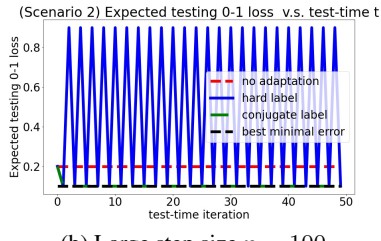

(a) Small step size $\eta = 1$           (b) Large step size $\eta = 100$

Figure 1: Expected $0-1$ loss vs. test-time iteration of GD. GD with hard labels under square loss (blue solid line) can not converge to the class mean $\mu$ in direction, while GD with conjugate labels under square loss (green dash dot line) learns an $\epsilon$-optimal predictor. Here, "no-adaptation" means simply predicting according to the initial model without any updates. The detailed setup is described in Appendix B.

## 5 CONVERGENCE RESULTS OF GD WITH PSEUDO-LABELS

Recall that we have $\ell^{\text{self}}(w; x) = \psi(w^\top x)$ for some scalar function $\psi(\cdot) : \mathbb{R} \to \mathbb{R}$ under the scenario of linear predictors. If $\psi(\cdot)$ is an even function, i.e., $\psi(u) = \psi(-u)$ for all $u \in \mathbb{R}$, then

$$\ell^{\text{self}}(w; x) = \psi(w^\top x) = \psi\left(yw^\top (\mu + \sigma\xi)\right) = \psi\left(w^\top (\mu + \sigma\xi)\right), \tag{16}$$

where the second equality uses $x = y(\mu + \sigma\xi)$ under Gaussian model, and the last equality uses the assumption that $\psi(\cdot)$ is an even function. We emphasize that the underlying algorithm itself does not have the knowledge of $\mu$, $\sigma$, or $\xi$, and the last expression simply arises from our analysis.

From (16), we know that the gradient is

$$\nabla\ell^{\text{self}}(w; x) = \nabla\psi(w^\top x) = \psi'\left(w^\top (\mu + \sigma\xi)\right)(\mu + \sigma\xi). \tag{17}$$

Hence, the dynamic of GD with pseudo-labels is

$$w_{t+1} = w_t - \eta\nabla\ell^{\text{self}}(w_t; x_t) = w_t - \eta\psi'\left(w_t^\top (\mu + \sigma\xi)\right)(\mu + \sigma\xi). \tag{18}$$

Now let us analyze the population dynamics, which means that we observe infinitely many unlabeled samples, so we can take expectation on the r.h.s. of (18). We get

$$w_{t+1} = w_t - \eta\mathbb{E}_\xi\left[\psi'\left(w_t^\top (\mu + \sigma\xi)\right)\right]\mu - \eta\mathbb{E}_\xi\left[\psi'\left(w_t^\top (\mu + \sigma\xi)\right)\sigma\xi\right] \tag{19}$$

$$= w_t - \eta\mathbb{E}_\xi\left[\psi'\left(w_t^\top (\mu + \sigma\xi)\right)\right]\mu - \eta\sigma^2\mathbb{E}_\xi\left[\psi''\left(w_t^\top (\mu + \sigma\xi)\right)\right]w_t$$

$$= \left(1 - \eta\sigma^2\mathbb{E}_\xi\left[\psi''\left(w_t^\top (\mu + \sigma\xi)\right)\right]\right)w_t - \eta\mathbb{E}_\xi\left[\psi'\left(w_t^\top (\mu + \sigma\xi)\right)\right]\mu, \tag{20}$$

where the second to last equality uses Stein's identity (Stein, 1981): for any function $\psi: \mathbb{R}^d \to \mathbb{R}$ and $\xi \sim \mathcal{N}(0, I_d)$, it holds that $\mathbb{E}_\xi[\xi\psi(\xi)] = \mathbb{E}_\xi[\nabla_\xi\psi(\xi)]$.

Denote $a_t := \langle w_t, \mu\rangle$ the dynamic of the component of $w_t$ along $\mu$. Given the dynamic (20), it is clear that the component along $\mu$ evolves as:

$$a_{t+1} = \left(1 - \eta\sigma^2\mathbb{E}_\xi\left[\psi''\left(w_t^\top (\mu + \sigma\xi)\right)\right]\right)a_t - \eta\mathbb{E}_\xi\left[\psi'\left(w_t^\top (\mu + \sigma\xi)\right)\right]\|\mu\|^2. \tag{21}$$

On the other hand, denote $b_t := \|[w_t[2], \ldots, w_t[d]]^\top\|$ the size of the component orthogonal to $\mu$. Then, its population dynamic evolves as:

$$b_{t+1} = \left|1 - \eta\sigma^2\mathbb{E}_\xi\left[\psi''\left(w_t^\top (\mu + \sigma\xi)\right)\right]\right|b_t. \tag{22}$$

We further define the ratio $r_t := \frac{a_t}{b_t}$. By (21) and (22), we have

$$r_{t+1} = \text{sign}\left(1 - \eta\sigma^2\mathbb{E}_\xi\left[\psi''\left(w_t^\top (\mu + \sigma\xi)\right)\right]\right)r_t + \frac{\eta\mathbb{E}_\xi\left[-\psi'\left(w_t^\top (\mu + \sigma\xi)\right)\right]\|\mu\|^2}{\left|1 - \eta\sigma^2\mathbb{E}_\xi\left[\psi''\left(w_t^\top (\mu + \sigma\xi)\right)\right]\right|b_t}. \tag{23}$$

It turns out that $\cos\left(\frac{w_t}{\|w_t\|}, \frac{\mu}{\|\mu\|}\right)$ is an increasing function of $r_t$, Indeed,

$$\cos\left(\frac{w_t}{\|w_t\|}, \frac{\mu}{\|\mu\|}\right) = \frac{\langle w_t, \mu\rangle}{\|w_t\|\|\mu\|} = \frac{\langle w_t, \mu\rangle}{b_t\sqrt{\|\mu\|^2 + \langle w_t, \mu\rangle^2/b_t^2}} = \text{sign}(r_t)\frac{1}{\sqrt{1 + \|\mu\|^2/r_t^2}}, \tag{24}$$

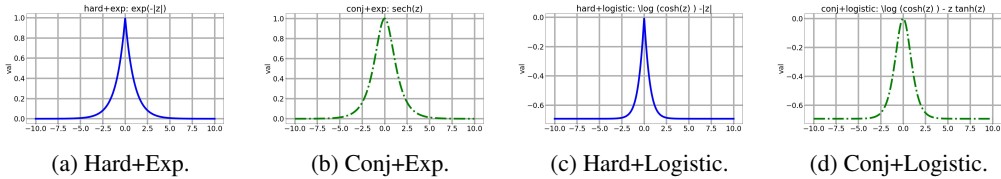

|  (a) Hard+Exp. | (b) Conj+Exp. | (c) Hard+Logistic. | (d) Conj+Logistic. |

Figure 2: Plots of some self-training loss functions that satisfy the set of properties ♣.

where we used $\|w_t\| = \sqrt{(w_t^\top \mu/\|\mu\|)^2 + b_t^2}$. A successful recovery (cos $\to 1$) means that we would like $r_t \to \infty$.

In the rest of this paper, we will use the notations $\diamond + \heartsuit$ or $GD + \diamond + \heartsuit$, where $\diamond = \{\text{conj}, \text{hard}\}$ and $\heartsuit = \{\text{square}, \text{logistic}, \text{exp}\}$ for brevity. For example, $\text{hard} + \text{exp}$ represents the self-training loss based on hard labels under exponential loss, i.e., $\ell^{\text{hard}}(w; x) = \exp(-|w^\top x|)$, while $GD + \text{conj} + \text{square}$ stands for GD with conjugate labels under square loss in test-time adaptation.

## 5.1 (EXPONENTIAL-RATE CONVERGENCE) $GD + \text{conj} + \text{square}$

**Proposition 2.** *($GD + \text{conj} + \text{square}$) The ratio of the projection onto $\mu$ to the size of the orthogonal component grows as*

$$r_{t+1} = r_1 \left(1 + \frac{\eta\|\mu\|^2}{1 + \eta\sigma^2}\right)^t.$$

*Furthermore, GD learns an $\epsilon$-optimal predictor after $t \geq \frac{1}{2} \frac{\log(\|\mu\|^2/(\epsilon r_1^2))}{\log(1+\eta\|\mu\|^2/(1+\eta\sigma^2))}$ iterations.*

*Proof.* For $GD + \text{conj} + \text{square}$, the self-training loss is $\ell^{\text{conj}}(w; x) = -\frac{1}{2}(w^\top x)^2$ from (6). Hence, $\psi(\cdot) = -\frac{1}{2}(\cdot)^2$ in (16); moreover, $\psi'(\cdot) = -(\cdot)$ and $\psi''(\cdot) = -1$ in (23). Therefore, we have $\mathbb{E}_\xi\left[-\psi'\left(w_t^\top(\mu + \sigma\xi)\right)\right] = \mathbb{E}_\xi\left[w_t^\top(\mu + \sigma\xi)\right] = w_t^\top\mu$ since $\mathbb{E}_\xi[w_t^\top\xi] = 0$, and we also have $\mathbb{E}_\xi\left[\psi''\left(w_t^\top(\mu + \sigma\xi)\right)\right] = \mathbb{E}_\xi[-1] = -1$ in (23).

Consequently, the dynamic of the ratio is

$$r_{t+1} = r_t + \frac{\eta w_t^\top \mu\|\mu\|^2}{(1 + \eta\sigma^2)b_t} = r_t\left(1 + \frac{\eta\|\mu\|^2}{1 + \eta\sigma^2}\right) = r_1\left(1 + \frac{\eta\|\mu\|^2}{1 + \eta\sigma^2}\right)^t. \tag{25}$$

From (24) and (25), the cosine between $w_t$ and $\mu$ is positive and increasing, given the initial condition $a_1 > 0$ (or equivalently, $r_1 > 0$). Hence, Condition 1 on (8) holds for all $t$. By using (24), we see that to get an $\epsilon$-optimal predictor at test time $t$, we need to satisfy $\|\mu\|^2 / \left(r_1^2\left(1 + \frac{\eta\|\mu\|^2}{1+\eta\sigma^2}\right)^{2t}\right) \leq \epsilon$.

Simple calculation shows that $t \geq \frac{1}{2} \frac{\log(\|\mu\|^2/(\epsilon r_1^2))}{\log(1+\eta\|\mu\|^2/(1+\eta\sigma^2))}$.

$\square$

Proposition 1 and 2 together provably show a performance gap between $GD + \text{conj} + \text{square}$ and $GD + \text{hard} + \text{square}$. Using conjugate labels, GD converges to the class mean $\mu$ in direction exponentially fast, while GD with hard labels fails in this task.

## 5.2 $\log(t)$-RATE CONVERGENCE OF GD

In this subsection, we consider self-training loss functions, $\ell^{\text{self}}(w; x) = \psi(w^\top x)$, that satisfy the following set of properties ♣ with parameter $(L, a_{\min})$: (i) Even: $\psi(-a) = \psi(a)$ for all $a \in \mathbb{R}$. (ii) There exists $0 < L < \infty$ such that $-\psi'(a) \geq e^{-La}$ for all $a \geq a_{\min}$.

**Lemma 1.** *The following self-training loss functions $\ell^{\text{self}}(w; x) = \psi(w^\top x)$ satisfy ♣. More precisely, we have:*

*1. $\text{hard} + \text{exp}$: $\psi(u) = \exp(-|u|)$ satisfies ♣ with $(L = 1, a_{\min} = 0)$.*

2. hard + logistic: $\psi(u) = \log\left(\cosh\left(u\right)\right) - |u|$ *satisfies* ♣ *with* $(L = 2, a_{\min} = 0)$.

3. conj + exp: $\psi(u) = \text{sech}(u)$ *satisfies* ♣ *with* $(L = 1, a_{\min} = 0.75)$.

4. conj + logistic: $\psi(u) = \log\left(\cosh\left(u\right)\right) - \tanh\left(u\right)u$ *satisfies* ♣ *with* $(L = 2, a_{\min} = 0.5)$.

The proof of Lemma 1 is available in Appendix C. Figure 2 plots the self-training losses listed in Lemma 1. From the figure, one might find that Property ♣ is evident for these self-training losses.

We will also need the following supporting lemma to get a convergence rate.

**Lemma 2.** *Consider the dynamic:* $r_{t+1} \geq r_t + ce^{-Lr_t}$, *for some* $L > 0$ *and* $c \geq 0$. *Suppose that initially* $r_1 > 0$. *Then,* $r_{t-\tau_*} \geq \frac{1}{2L}\log c(t-1)$, *for all* $t > \tau_*$, *where* $\tau_* = 0$ *if* $\nu \leq e^{L\nu}, \forall \nu \geq 0$; *otherwise,* $\tau_* = \nu_*^2(L)/c$, *where* $\nu_*(L)$ *is the unique fixed point of* $\nu_* = e^{L\nu_*}$ *if it exits.*

*Proof.* From the dynamic, it is clear that $r_{t+1} \geq r_t$ since $c \geq 0$. Then,

$$e^{Lr_{t+1}}r_{t+1} \geq e^{Lr_{t+1}}r_t + ce^{L(r_{t+1}-r_t)} \geq e^{Lr_t}r_t + c \geq e^{Lr_0}r_0 + ct \geq ct, \tag{26}$$

where the last step follows from unrolling the recursion $t$ times.

We first analyze the case that $r_t \leq e^{Lr_t}$. Since $r_t \leq e^{Lr_t}$, we have $e^{2Lr_t} \geq c(t-1)$ from (26). Hence, $r_t \geq \frac{1}{2L}\log c(t-1)$.

Now let us switch to the case that $r_t \geq e^{Lr_t}$. Let $\nu_*(L)$ the unique point of $\nu_*$ such that $\nu_* = e^{L\nu_*}$. If $r_t \leq \nu_*(L)$, then $r_t \geq e^{Lr_t}$. Hence, we have $r_t^2 \geq r_t e^{Lr_t} \overset{(26)}{\geq} c(t-1)$. So $r_t \geq \sqrt{c(t-1)}$. Note this possibility cannot happen more than $\tau_* := \nu_*^2(L)/c$ times, since we need $r_t \leq r_*$ to stay in this regime. So eventually we get out of this regime after a constant number $\tau_*$ iterations. $\square$

Now we are ready to state another main result in this paper. Proposition 3 below shows a $\log(t)$-convergence rate of GD with pseudo-labels in the noiseless setting $\sigma^2 = 0$ if the underlying self-training loss function satisfies ♣. The gap between the exponential rate of GD with conjugate labels using square loss shown in Proposition 2 and the logarithmic rate in Proposition 3 suggests that the performance of GD in test-time adaptation also crucially depends on the choice of loss functions, in addition to the choice of pseudo-labels.

**Proposition 3.** *(Noiseless setting) Apply GD to minimizing* $\ell^{\text{self}}(w; x) = \psi(w^\top x)$, *where* $\psi(\cdot)$ *satisfies* ♣. *If the initial point satisfies* $a_1 > a_{\min}$, *then the ratio of* $w_t's$ *component along* $\mu$ *to the size of its orthogonal component to* $\mu$ *at test time* $t$, *i.e.,* $r_t$ *in (23), satisfies*

$$r_{t-\tau_*} = \Omega\left(\frac{1}{Lb_1}\log\left(\frac{\eta\|\mu\|^2}{b_1}t\right)\right), \text{ for all } t > \tau_*,$$

*where* $\tau_*$ *is a constant defined in Lemma 2.*

*Proof.* From (19) or (22), we know that the size of the orthogonal component does not change throughout the iterations when $\sigma^2 = 0$, i.e., $b_{t+1} = b_t, \forall t$. On the other hand, the component along $\mu$ in the noiseless setting has the dynamic,

$$a_{t+1} \overset{(21)}{=} a_t + \eta\left(-\psi'\left(a_t\right)\right)\|\mu\|^2 \geq a_t + \eta e^{-La_t}\|\mu\|^2, \forall a_t \geq a_{\min}, \tag{27}$$

where we recall $a_t := \langle w_t, \mu \rangle$ and the inequality uses the property regarding $-\psi'(\cdot)$ as stated in ♣. It is noted that (27) implies that $a_t$ is non-decreasing, and hence the condition about the initial point, i.e., $a_1 \geq a_{\min}$, guarantees $a_t \geq a_{\min}$ for all test time $t$.

By using the above results, we deduce that the dynamic of the ratio $r_t := \frac{a_t}{b_t}$ satisfies $r_{t+1} \geq r_t + \frac{\eta e^{-La_t}\|\mu\|^2}{b_1} = r_t + \frac{\eta e^{-Lr_t b_1}\|\mu\|^2}{b_1}$, where we used that $b_{t+1} = b_t = b_1, \forall t$. Invoking Lemma 2 leads to the result.

$\square$

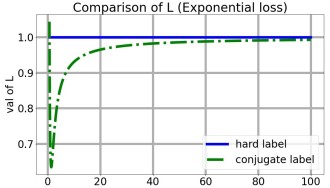 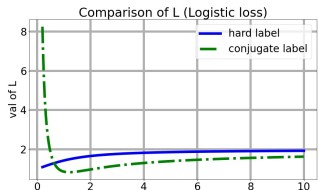

Figure 3: We plot $L(z) := \frac{\log(-\psi'(z))}{z}$ vs. $z$, where $\psi'(\cdot)$ is the first derivative of the underlying self-training loss. Left: $L(z)$ vs. $z$ of $\mathrm{hard} + \exp$ and $\mathrm{conj} + \exp$. Right: $L(z)$ vs. $z$ of $\mathrm{hard} + \mathrm{logistic}$ and $\mathrm{conj} + \mathrm{logistic}$.

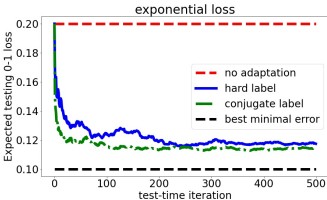 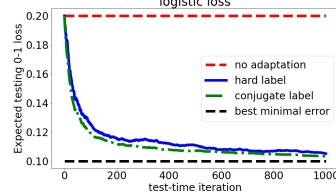

Figure 4: Expected 0-1 loss $\Phi\left(\frac{\mu^\top w_t}{\sigma \|w_t\|}\right)$ vs. test-time $t$. Left: $\mathrm{GD} + \mathrm{hard} + \exp$ and $\mathrm{GD} + \mathrm{conj} + \exp$. Right: $\mathrm{GD} + \mathrm{hard} + \mathrm{logistic}$ and $\mathrm{GD} + \mathrm{conj} + \mathrm{logistic}$. Here "best minimal error" is $\Phi\left(\frac{\|\mu\|}{\sigma}\right)$ (recall the discussion in Section 3). Both figures show that GD with conjugate labels outperforms GD with hard labels.

Proposition 3 implies that GD for minimizing a self-training loss with a smaller constant $L$ can result in a faster growth of the ratio $r$ and consequently a faster convergence rate. Recall the definition of $L$ in Property ♣: a smaller constant $L$ means that the (minus) derivative $-\psi'(\cdot)$ of the self-training loss has a heavier tail. We therefore compare the tails of the self-training loss functions by plotting $L(z) := \frac{\log(-\psi'(z))}{z}$ of each on Figure 3, which shows that there exists a threshold $z_{\min}$ such that for all $z \geq z_{\min}$, the number $L(z)$ that corresponds to the loss function with the conjugate label is smaller than that of the hard label. This implies that the self-training loss derived from conjugate labels can have a smaller constant $L$ (for a *finite $z$*) compared to that of hard labels, which in turn might hint at a faster convergence of $\mathrm{GD} + \mathrm{conj}$ compared to $\mathrm{GD} + \mathrm{hard}$ for exponential loss and logistic loss. Figure 4 shows the experimental results under Gaussian model, where GD uses a received mini-batch of samples to conduct the update at each test time. The detailed setup is available in Appendix B. We find that GD with conjugate labels dominates GD with hard labels empirically, which is aligned with our theoretical result. It is noted that for the case of exponential loss, Goyal et al. (2022) report a similar experimental result under Gaussian model — $\mathrm{GD} + \mathrm{conj} + \exp$ outperforms $\mathrm{GD} + \mathrm{hard} + \exp$.

## 6 LIMITATIONS AND OUTLOOKS

In this paper, we analyze GD with hard and conjugate pseudo-labels for test-time adaptation under different loss functions. We study the performance of each of them under a binary classification framework, identify a scenario when GD with hard labels cannot converge to an $\epsilon$-optimal predictor for any small $\epsilon$ while GD with conjugate labels does, and obtain some convergence results of GD with pseudo-labels. However, there are still many directions worth exploring. First of all, while our current analysis in the binary classification setting might be viewed as a first step towards systematically studying GD with pseudo-labels, analyzing GD with pseudo-labels in multi-class classification is left open in this work and could be a potential direction. Second, while analyzing the population dynamics has already given us some insights about GD with pseudo labels, it might be useful to study their finite-sample dynamics. Third, theoretically understanding GD with other pseudo-labels or combined with other domain adaptation techniques like ensembling (e.g., Wortsman et al. (2022)) or others (e.g., Li et al. (2019); Schneider et al. (2020); Eastwood et al. (2022)) might be promising. Finally, analyzing momentum methods (e.g., Nesterov (2013); Wibisono et al. (2016); Wang & Abernethy (2018); Wang et al. (2022a; 2021b;c)) with pseudo-labels is another interesting direction, and one of the open questions is whether they enjoy provable guarantees of faster test-time adaptation compared to GD. Overall, we believe that the connection between optimization, domain adaptation, and machine learning under distribution shifts can be strengthened.

ACKNOWLEDGMENTS

The authors appreciate Shikhar Jaiswal spotting a minor error in our previous version of the proof of Proposition 1, which has been corrected in this version. The authors thank the constructive feedback from the reviewers and comments from Sachin Goyal, which helps improve the quality of this paper. The authors also thank Chi-Heng Lin for valuable discussions.

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

## A   DERIVATIONS OF CONJUGATE LABELS AND THE ASSOCIATED SELF-TRAINING LOSSES ON TABLE 1

**1. (Square loss):** Square loss $\ell(w; (x, y)) := \frac{1}{2}(y - w^\top x)^2$ is in the form of (2), where $f(\cdot) = \frac{1}{2}(\cdot)^2 : \mathbb{R} \to \mathbb{R}^+$. Substituting $f(\cdot) = \frac{1}{2}(\cdot)^2$ and $h(w) = w^\top x$ into (4) and (5), we get

$$y_w^{\mathrm{conj}}(x) = w^\top x, \quad \text{and} \quad \ell^{\mathrm{conj}}(w; x) = -\frac{1}{2}(w^\top x)^2. \tag{28}$$

On the other hand, let $y \leftarrow \mathrm{sign}(w^\top x)$. we have

$$y_w^{\mathrm{hard}}(x) = \mathrm{sign}(w^\top x), \quad \text{and} \quad \ell^{\mathrm{hard}}(w; x) = \frac{1}{2}\left(\mathrm{sign}\left(w^\top x\right) - w^\top x\right)^2. \tag{29}$$

**2. (Logistic loss):** Recall that logistic regression predicts $P(\hat{y} = 1) = \frac{\exp(w^\top x)}{1+\exp(w^\top x)}$ and $P(\hat{y} = 0) = 1 - P(\hat{y} = 1)$, and the loss function is:

$$
\begin{aligned}
\ell^{\text{logit}}(w; (x, \hat{y})) &:= - \left( \hat{y} \log(P(\hat{y} = 1)) + (1 - \hat{y}) \log\left( P(y = 0) \right) \right) \\
&:= \log\left( 1 + \exp(w^\top x) \right) - \hat{y}(w^\top x),
\end{aligned}
\tag{30}
$$

where $\hat{y} = \{0, 1\}$. Let $y = 2\hat{y} - 1 \in \{-1, 1\}$. Then, substituting $\hat{y} = \frac{1}{2} + \frac{y}{2}$ back into (30) and using the equation $\cosh(z) = \frac{\exp(z) + \exp(-z)}{2}$ for any $z \in \mathbb{R}$, we obtain an equivalent objective:

$$
\begin{aligned}
\ell^{\text{logit}}(w; (x, y)) &= \log(1 + \exp(w^\top x)) - \hat{y}(w^\top x) \\
&= \log(1 + \exp(w^\top x)) - \left( \frac{1}{2} + \frac{y}{2} \right)(w^\top x) \\
&= \log\left( \exp\left( \frac{w^\top x}{2} \right) + \exp\left( -\frac{w^\top x}{2} \right) \right) - y \frac{w^\top x}{2} \\
&= \log\left( \cosh\left( \frac{w^\top x}{2} \right) \right) - y \frac{w^\top x}{2} + \log 2.
\end{aligned}
\tag{31}
$$

Now by renaming $\frac{w}{2} \leftarrow w$, we get

$$
\ell^{\text{logit}}(w; (x, y)) = \log\left( \cosh\left( w^\top x \right) \right) - y w^\top x + C,
\tag{32}
$$

where the last term is a constant and can be dropped without affecting the training.

Observe that (32) is in the form of (2), where $f(\cdot) = \log\left( \cosh\left( \cdot \right) \right)$ and $h_w(x) = w^\top x$. Using (4) and (5), we get

$$
y_w^{\text{conj}}(x) = \tanh\left( w^\top x \right), \quad \text{and} \quad \ell^{\text{conj}}(w; x) = \log\left( \cosh\left( w^\top x \right) \right) - \tanh\left( w^\top x \right) w^\top x.
\tag{33}
$$

On the other hand, let $y \leftarrow \text{sign}(w^\top x)$ in (32). we have

$$
y_w^{\text{hard}}(x) = \text{sign}(w^\top x), \quad \text{and} \quad \ell^{\text{hard}}(w; x) = \log\left( \cosh\left( w^\top x \right) \right) - |w^\top x|.
\tag{34}
$$

**3. (Exponential loss):** Recall that exponential loss is $\ell^{\text{exp}}(w; (x, y)) := \exp(-y h_w(x)) = \exp(-y w^\top x)$, where $y = \{+1, -1\}$, which can be rewritten as

$$
\begin{aligned}
\ell^{\text{exp}}(w; (x, y)) &= \frac{1}{2} \left( \exp(w^\top x) + \exp(-w^\top x) \right) - \frac{1}{2} y \left( \exp(w^\top x) - \exp(-w^\top x) \right), \\
&= \cosh(w^\top x) - y \sinh(w^\top x).
\end{aligned}
\tag{35}
$$

The above function is in an *expanded* conjugate form (Goyal et al., 2022):

$$
f(h_w(x)) - y g(h_w(x)),
$$

where $f(\cdot) = \cosh(\cdot)$, $g(\cdot) = \sinh(\cdot)$, and $h_w(x) = w^\top x$. Let $h_* \leftarrow \arg\min_h f(h) - y g(h)$. Then, $h_*$ satisfies $\nabla f(h_*) = \nabla g(h_*) y$. Goyal et al. (2022) define the conjugate label $y_w^{\text{conj}}(x)$ via the equality

$$
\nabla f(h_w(x)) = \nabla g(h_w(x)) y_w^{\text{conj}}(x)
$$

for this case. Therefore, we have $y_w^{\text{conj}}(x) = \tanh(w^\top x)$. By substituting $y \leftarrow y_w^{\text{conj}}(x)$ in (35), we get the self-training loss function using the conjugate label: $\ell^{\text{conj}}(w) = \text{sech}(w^\top x)$. To conclude, we have:

$$
y_w^{\text{conj}}(x) = \tanh(w^\top x), \quad \text{and} \quad \ell^{\text{conj}}(w; x) = \text{sech}(w^\top x).
\tag{36}
$$

On the other hand, let $y \leftarrow \text{sign}(w^\top x)$ in $\ell^{\text{exp}}(w; (x, y)) := \exp(-y h_w(x))$, we have

$$
y_w^{\text{hard}}(x) = \text{sign}(w^\top x), \quad \text{and} \quad \ell^{\text{hard}}(w; x) = \exp(-|w^\top x|).
\tag{37}
$$

## B    SETUP OF THE SIMULATION IN FIGURE 1 AND FIGURE 4

Below we describe how to reproduce Figure 1 and Figure 4. We first specify the mean and covariance $\mu_{\mathcal{S}}, \mu_{\mathcal{T}}, \Sigma_{\mathcal{S}} = \sigma_{\mathcal{S}} I_d, \Sigma_{\mathcal{T}} = \sigma_{\mathcal{T}} I_d$ as follows, where the subscript $\mathcal{S}$ stands for the source domain, and the subscript $\mathcal{T}$ is the target domain.

We set $\mu_{\mathcal{S}} = e_1$ and then set set $\mu_{\mathcal{T}}[1] = 0.6567$, and the remaining elements of $\mu_{\mathcal{T}}$ is set randomly from a normal distribution and were normalized to ensure that $\mu_{\mathcal{T}}$ is a unit norm vector. Then, we set $\sigma_{\mathcal{T}} = 0.6567/0.8416$. This way we have $\frac{\mu_{\mathcal{T}}^{\top} \mu_{\mathcal{S}}}{\sigma_{\mathcal{T}} \|\mu_{\mathcal{S}}\|} = 0.8416$ so that $\Phi\left(\frac{\mu_{\mathcal{T}}^{\top} \mu_{\mathcal{S}}}{\sigma_{\mathcal{T}} \|\mu_{\mathcal{S}}\|}\right) = \Phi(0.8416) = 0.2$, i.e., the initial model $w_1 = w_{\mathcal{S}}$ has $20\%$ expected $0-1$ loss in the new domain $\mathcal{T}$. Also, the best minimal error in the new domain $\mathcal{T}$ is $\Phi\left(\frac{\|\mu_{\mathcal{T}}\|}{\sigma_{\mathcal{T}}}\right) = \Phi\left(\frac{1}{0.6567/0.8416}\right) = 0.1$.

In the simulation result depicted in Figure 1, a sample of $(x = \mu)$ arrives when the test time $t$ is an odd number and a sample of $(x = -\mu)$ arrives when the test time $t$ is an even number. Note that the algorithms do not know the labels.

In the simulation result depicted in Figure 4, we consider the setting of noisy data, i.e., $x_t \in \mathbb{R}^d$ is sampled as $x_t \sim \mathcal{N}(\mu_{\mathcal{T}}, \sigma_{\mathcal{T}}^2 I_d)$ instead of $x_t = y\mu_{\mathcal{T}}$. We search the step size $\eta$ over the grid $\{10^{-3}, 5 \times 10^{-3}, 10^{-2}, 5 \times 10^{-2}, 10^{-1}, 5 \times 10^{-1}, 10^0, 5 \times 10^0, 10^1, 5 \times 10^1, 10^2\}$ for each $\mathrm{GD} + \mathrm{hard} + \exp$, $\mathrm{GD} + \mathrm{conj} + \exp$, $\mathrm{GD} + \mathrm{hard} + \mathrm{logistic}$, or $\mathrm{GD} + \mathrm{conj} + \mathrm{logistic}$, and report the best result of each one.

## C    PROOF OF LEMMA 1

**Lemma 1:**    *The following self-training loss functions $\ell^{\mathrm{self}}(w; x) = \psi(w^{\top} x)$ satisfy the set of properties ♣. More precisely, we have*

1. $\mathrm{hard} + \exp$: $\psi(u) = \exp(-|u|)$ *satisfies* ♣ *with* $(L = 1, a_{\min} = 0)$.
2. $\mathrm{hard} + \mathrm{logistic}$: $\psi(u) = \log(\cosh(u)) - |u|$ *satisfies* ♣ *with* $(L = 2, a_{\min} = 0)$.
3. $\mathrm{conj} + \exp$: $\psi(u) = \mathrm{sech}(u)$ *satisfies* ♣ *with* $(L = 1, a_{\min} = 0.75)$.
4. $\mathrm{conj} + \mathrm{logistic}$: $\psi(u) = \log(\cosh(u)) - \tanh(u) u$ *satisfies* ♣ *with* $(L = 2, a_{\min} = 0.5)$.

*Proof.*

- For $\mathrm{hard} + \exp$, we have $\psi(u) = \exp(-|u|)$, $\psi'(u) = -\mathrm{sign}(u) \exp(-|u|)$, and $\psi''(u) = \exp(-|u|) + \delta_0(u)$.

  It is evident that $\psi(u) = \exp(-|u|)$ is an even function and that it is differentiable everywhere except at the origin. We also have $|-\psi'(u)| \leq 1$ and $-\psi'(u) \geq \exp(-u)$ for all $u \geq 0$. We conclude that $\psi(u) = \exp(-|u|)$ satisfies ♣ with parameter $(L = 1, a_{\min} = 0)$.

- For $\mathrm{hard} + \mathrm{logistic}$, we have $\psi(u) = \log(\cosh(u)) - |u|$, $\psi'(u) = \tanh(u) - \mathrm{sign}(u)$, and $\psi''(u) = \mathrm{sech}^2(u) - \delta_0(u)$.

  It is evident that $\psi(u) = \log(\cosh(u)) - |u|$ is an even function and that it is differentiable everywhere except at the origin. We also have $|-\psi'(u)| \leq 1$. Furthermore,

  $$\tanh(u) - 1 = \frac{\exp(u) - \exp(-u)}{\exp(u) + \exp(-u)} - 1 = -\frac{2\exp(-u)}{\exp(u) + \exp(-u)}.$$

  Hence, for $u > 0$, $-\phi'(u) = 1 - \tanh(u) = \frac{2\exp(-u)}{\exp(u)+\exp(-u)} \geq \exp(-2u)$, since

  $$\frac{2\exp(-u)}{\exp(u) + \exp(-u)} \geq \exp(-2u) \iff 2\exp(-u) \geq \exp(-u) + \exp(-3u),$$

  and the later is evident for $u \geq 0$.

  We conclude that $\psi(u) = \log(\cosh(u)) - |u|$ satisfies ♣ with parameter $(L = 2, a_{\min} = 0)$.

- For conj + exp, we have $\psi(u) = \mathrm{sech}(u)$, $\psi'(u) = -\tanh(u)\mathrm{sech}(u)$, and $\psi''(u) = -\mathrm{sech}(u)^3 + \tanh^2(u)\mathrm{sech}(u)$.

  It is evident that $\psi(u) = \mathrm{sech}(u)$ is an even function and that it is differentiable everywhere. We also have $|-\psi'(u)| \leq 1$, as $|\tanh(u)| \leq 1$ and $\mathrm{sech}(u) \leq 1$.

  Note that $-\psi'(u) = \tanh(u)\mathrm{sech}(u) = \frac{2(\exp(u)-\exp(-u))}{(\exp(u)+\exp(-u))^2}$. Moreover,

  $$\frac{2(\exp(u) - \exp(-u))}{(\exp(u) + \exp(-u))^2} \geq \exp(-u) \iff 2(\exp(2u) - 1) \geq \exp(2u) + 2 + \exp(-2u)$$
  $$\iff \exp(2u) \geq \exp(-2u) + 4, \tag{38}$$

  which holds when $u \geq 0.75$. That is, $-\psi'(u) \geq \exp(-u)$ for all $u \geq 0.75$.

  We conclude that $\psi(u) = \mathrm{sech}(u)$ satisfies ♣ with parameter $(L = 1, a_{\min} = 0.75)$.

- For conj + logistic, we have $\psi(u) = \log(\cosh(u)) - \tanh(u)u$, $\psi'(u) = -\mathrm{sech}^2(u)u$, and $\psi''(u) = -\mathrm{sech}(u)^2 + 2u\tanh(u)\mathrm{sech}^2(u)$.

  It is evident that $\psi(u) = \log(\cosh(u)) - \tanh(u)u$ is an even function and that it is differentiable everywhere. We also have $|-\psi'(u)| = \left|\frac{4u}{(\exp(u)+\exp(-u))^2}\right| \leq 1$.

  Note that $-\psi'(u) = \mathrm{sech}^2(u)u = \frac{4u}{(\exp(u)+\exp(-u))^2}$. Moreover,

  $$\frac{4u}{(\exp(u) + \exp(-u))^2} \geq \exp(-2u) \iff 4u \geq 1 + 2\exp(-2u) + \exp(-4u), \tag{39}$$

  which holds when $u \geq 0.5$. That is, $-\psi'(u) \geq \exp(-2u)$ for all $u \geq 0.5$.

  We conclude that $\psi(u) = \log(\cosh(u)) - \tanh(u)u$ satisfies ♣ with parameter $(L = 2, a_{\min} = 0.5)$.

  $\square$

