# OpenReview forum: "Towards Understanding GD with Hard and Conjugate Pseudo-labels for Test-Time Adaptation"
_ICLR.cc/2023/Conference — ICLR 2023 poster_

### Official Review · Reviewer_w2JR · 2022-10-17

**Confidence:** 3
**Correctness:** 4
**Technical Novelty And Significance:** 3
**Empirical Novelty And Significance:** 3
**Recommendation:** 6

**Clarity, Quality, Novelty And Reproducibility:**

Clarity: The paper is well-organized and clearly written.

Quality: Technically solid paper.

Novelty: The paper makes non-trivial advances over the current state-of-the-art.

Reproducibility: Key details are sufficiently well-described for competent researchers to confidently reproduce the main results.

**Strength And Weaknesses:**

Strength：
- The paper is well-organized and clearly written, which is easy to follow.
- The theoretical work of this paper is sufficient, which improves the value of the paper.
- The paper provides insights into when and why GD with hard labels or conjugate labels works in test-time adaptation.

Weakness：
- The original conjugate pseudo-labels[A] perform results across different domain adaptation. The author neglect the explaination of this scenarios. Is the convergence analysis general for diverse domain with large gap?
- My concern is the assumption in equation (8). The authors have not provided the value range of $\epsilon$. Although the authors mention,”... as it means that the
source model is better than the random guessing in the new domain.”, how to illustrate the special case of right angle when $\epsilon=1$？
- The convergence of GD was analyzed in section 5. What is the main obstacle to investigating convergence according to the idea proposed in this paper？ Moreover, in section 5.2, does $log(t)$-rate convergence boost compared to the convergence rate of related pseudo labels？

[A] Sachin Goyal, Mingjie Sun, Aditi Raghunathan, and Zico Kolter. Test-Time Adaptation via Conjugate Pseudo-labels. NeurIPS, 2022.



**Summary Of The Paper:**

This paper presents theoretical insight to explain GD with hard and conjugate labels for a binary classification problem.This paper shows that for square loss, GD with conjugate labels converges to a solution that minimizes the testing 0-1 loss under a Gaussian model, while GD with hard pseudo-labels breaks down. Besides，the authors discuss them under different loss functions.

**Summary Of The Review:**

Please see Q1 and Q2.

---

> ### Author Response · Authors · 2022-11-12
> **Thanks for the comments and the positive feedback.**
>
> Thanks for the comments and the positive feedback. Please kindly find our response to each of the questions below.
>
> __*1. The original conjugate pseudo-labels[A] perform results across different domain adaptation. The author neglect the explanation of this scenarios. Is the convergence analysis general for diverse domain with large gap?*__
>
> Thanks for raising this question. The work of Goyal et al. (2022) shows promising empirical results of conjugate labels across many domain adaptation datasets. The ways to construct these datasets are quite different, e.g., novel camera viewpoints or novel image backgrounds. Theoretically analyzing the behavior of gradient descent with pseudo-labels on these datasets/scenarios is beyond the scope of this paper.
>
> __*2. My concern is the assumption in equation (8). The authors have not provided the value range of $\epsilon$. Although the authors mention, as it means that the source model is better than the random guessing in the new domain. how to illustrate the special case of right angle when $\epsilon=1$?*__
>
> Allow us to replicate (8) here for the reader's convenience:
> $$\textbf{Condition 1:} \quad  \left \langle w , \frac{\mu}{ \| \mu \|} \right \rangle = w[1] > 0
>  \quad \text{and} \quad  \textbf{Condition 2:} \quad  \cos^2\left(\frac{w}{\|w\|}, \frac{\mu}{\| \mu\|} \right) \geq 1 -\epsilon.
> $$
>
>
> We first would like to clarify that the two conditions on (8) are NOT the assumptions, instead, they are regarding the definition of an $\epsilon$-optimal predictor. Namely, a model is an $\epsilon$-optimal predictor if it satisfies both conditions.
>
> When $\epsilon=1$, the r.h.s. of the inequality (of Condition 2) is $0$, and hence Condition 2 trivially holds for any $w$ and is not an interesting case.
> To approach the best accuracy,
> we would like the parameter $\epsilon$ to be as small as possible, i.e., $\epsilon \rightarrow 0 $, which can be seen from the discussion right below (8), where we state that the expected 0-1 loss of an $\epsilon$-optimal classifier is $\Phi\left( \frac{\| \mu \|}{ \sigma} \sqrt{1-\epsilon} \right)$. (Recall that $\Phi(v):= \frac{1}{\sqrt{2\pi}} \int_v^{\infty} \exp(-z^2/2) dz$ is the Gaussian error function.) We will emphasize this in the revised version.
>
> We assume that the source model satisfies Condition 1 on (8), but not Condition 2, i.e., the source model does not satisfy Condition 2 for any arbitrarily small $\epsilon$, which implies that the source model is not an $\epsilon$-optimal predictor for any arbitrarily small $\epsilon$ and needs to be adapted to the new domain.
>
> Thanks for raising this clarifying question, which helps improve the presentation of the current work.
>
> __*3.  The convergence of GD was analyzed in section 5. What is the main obstacle to investigating convergence according to the idea proposed in this paper? Moreover, in section 5.2, does $\log(t)$-rate convergence boost compared to the convergence rate of related pseudo labels?*__
>
> One main obstacle is that relevant theoretical results of GD with  pseudo-labels are sparse in the literature, and hence we have to derive the analysis in Section 4 and 5 from scratch.
> From our analysis, an important property of the problem that affects the convergence rates is the tail behavior of the (derivative of the) self-training loss function, e.g. see condition (ii) in the definition of the property $\clubsuit$ in Section 5.2. For the cases we analyzed in this paper we could control the tail behavior and prove convergence guarantees; however, extending our results to more general loss functions requires a good control on the tail behavior.
>
>
> The $\log(t)$ rate holds for
> (1) GD + exp loss with hard labels,
> (2) GD + exp loss with conjugate labels,
> (3) GD + logistic loss with hard labels, and
> (4) GD + logistic loss with conjugate labels,
> since Lemma 1 shows that the resulting self-training loss functions satisfy the assumption of Proposition 3.
> In other words,
> the log-$(t)$ rate holds for minimizing the
> self-training loss functions from the combinations of
> { conjugate, hard } + {  logistic loss, exp loss } via GD.
> On the other hand, GD + square loss with conjugate labels has an $\exp(t)$ rate,
> as we shown in Section 5.1, which is better than the $\log(t)$ rate.
> The difference of the convergence rates suggests that the performance of GD in test-time adaptation also crucially depend on the choice of loss functions, in addition to the choice of pseudo-labels.
>
> ==================================================================
>
> It would be appreciated if the reviewer can consider upgrading the score if we have provided satisfying answers. Or, it would also be appreciated if the reviewer can let us know whether we could address any parts in more detail. Thank you.

---

### Official Review · Reviewer_4HLp · 2022-10-24

**Confidence:** 4
**Correctness:** 4
**Technical Novelty And Significance:** 4
**Empirical Novelty And Significance:** 4
**Recommendation:** 6

**Clarity, Quality, Novelty And Reproducibility:**

The theoretical analysis and results are novel enough and the presentation is clear. However, the assumptions of the paper are a bit too strong, and the conclusions of the paper have limitations.

**Strength And Weaknesses:**

1. This paper focuses on binary classification problem, and the authors analyze three classical loss functions: square loss, logistic loss, and exponential loss. But binary cross-entropy loss has been widely studied in community, especially for training deep models. So, I am interested in the results of this loss function.
2. The assumptions adopted by the author seem to be somewhat strict, such as that the conjugate function must be an even function. We know that not all functions satisfy this constraint. If it is an odd function, does the conclusion in the paper still hold?
3. We find that the Gaussian models used in Section 3 and Section 5 are different. How much does this different choice affect the conclusion? If we still use the Gaussian model in Section 3, will the conclusions of Section 5 not hold?
4. In section 5, we observe that the normalized operation of the parameter w is ignored. Whether or not to use normalized has a large impact on the results, and the authors need to clarify this modification.
5. How to derive Equation 24 is not very clear.


**Summary Of The Paper:**

This paper is a theoretical analysis of a recent work [a]. They use the Gaussian model to explain and understand the gradient descent with hard-label and conjugate labels. For convenience analysis, they only consider the binary classification problem. The theoretical results and analysis are interesting.

[a] Test-Time Adaptation via Conjugate Pseudo-labels. NeurIPS, 2022.

**Summary Of The Review:**

This paper gives an interesting understanding of the GD with hard and conjugate labels for test-time training. I think this will bring new insights to the community.

---

> ### Author Response · Authors · 2022-11-12
> **Thanks for the comments and the positive feedback. (1/2)**
>
> Thanks for the comments and the positive feedback. Please kindly find our response to each of the comments/questions below.
>
> __*1. This paper focuses on binary classification problem, and the authors analyze three classical loss functions: square loss, logistic loss, and exponential loss. But binary cross-entropy loss has been widely studied in community, especially for training deep models. So, I am interested in the results of this loss function.*__
>
> Thanks for raising this point. We note that binary cross-entropy loss is equivalent to logistic loss. To see this,
> let us first recall the definition of the $K$-multi-class cross-entropy loss function.
> Denote $h_{W}(x)$, a $K$-dimensional vector consists of the predicted score of each class by a model $W$. Then, the $K$-class cross-entropy loss is
> $$\ell(W;(y,x))  := -\sum_{k=1}^K y[k] \log \left(  \frac{ \exp( h_W(x)[k] ) }{\sum_{k'} \exp( h_W(x)[k'] ) }   \right)
>  = \log\left( \sum_{k=1}^K  \exp( h_W(x)[k]) \right) - y^\top h_W(x),$$
> where $y$ is a $K$-dimensional one-hot encoding vector for the label of the sample $x$.
>
> Recall also the setting of logistic regression,
> which models a binary classification problem ($y = \\{1, 0\\}$).
> More precisely, it parametrizes the probability as
> $P(y=1|x) = \frac{\exp( w^\top x)}{  1 + \exp( w^\top x) }$
> and $P(y=0|x) = 1 - P(y=1|x)$.
> Then, the objective of logistic regression is
> $$
> \ell(w;(y,x)) := - \left( \ y \log ( P(y=1|x) )  + (1-y) \log \left( P(y=0|x) \right) \right)
> \ = \log \left(  1 + \exp( w^\top x ) \right) - y ( w^\top x).
> $$
> By way of the comparison, we see that logistic regression is equivalent to $K$-cross-entropy loss when $K=2$, where $W \in \mathbb{R}^{d \times 2} = [w, 0_d]$ and $h_W(x) = W^\top x$.
>
> It is also noted that the objective function of logistic regression
> $\log \left(  1 + \exp( w^\top x ) \right) - y ( w^\top x), y \in \\{0,1\\}$
> is equivalent to that of $\log ( \cosh(w^\top x) ) - y (w^\top x), y \in \\{-1,1\\}$ as presented in the paper; see Appendix A in our paper for more details on this equivalence.
>
> Therefore, our results also cover binary cross-entropy loss. We thank the reviewer for raising this point, and we are happy to emphasize it in the revised version.
>
> __*2. The assumptions adopted by the author seem to be somewhat strict, such as that the conjugate function must be an even function. We know that not all functions satisfy this constraint. If it is an odd function, does the conclusion in the paper still hold?*__
>
> We would like to clarify that the self-training loss function must be an even function is NOT an assumption, but rather a property that emerges from our analysis. As described in the paper, we analyze GD with pseudo-labels with the most common loss functions for the task of binary classification, i.e., square loss, logistic loss, and exponential loss. We found that when coupling these loss functions with hard pseudo-labels or conjugate pseudo-labels, the resulting self-training loss functions are even functions.
> That is, the
> self-training loss functions resulted from the combinations of
> { conjugate, hard } + {  square loss , logistic loss, exp loss } for the task of binary classification are all even.
> The self-training loss functions being even is a property that arises from coupling the common loss functions with pseudo-labels, not an assumption.
> Our analysis did use that the resulting self-training loss functions are even, but this is because we leverage the underlying structure of the resulting self-training losses to analyze the GD dynamics. Our conclusions and theoretical results are about minimizing these self-training loss functions via GD accordingly, and hence they do not cover the case when the self-training loss is an odd function, as this is irrelevant in our setting. But we think that identifying a scenario when the resulting self-training loss is odd
> and analyzing the GD dynamic in that case can be an interesting future direction.
>
> __*3. We find that the Gaussian models used in Section 3 and Section 5 are different. How much does this different choice affect the conclusion? If we still use the Gaussian model in Section 3, will the conclusions of Section 5 not hold?*__
>
> We would like to clarify that they are the same. Section 3 describes our theoretical framework, i.e. Gaussian model. Section 5 are about the convergence results under the theoretical framework introduced in Section 3.

---

> > ### Author Response · Authors · 2022-11-12
> > **Thanks for the comments and the positive feedback (2/2)**
> >
> > __*4. In section 5, we observe that the normalized operation of the parameter w is ignored. Whether or not to use normalized has a large impact on the results, and the authors need to clarify this modification.*__
> >
> > We would like to clarify that our analysis is about vanilla GD (without normalization), and hence we are not sure about what the reviewer means by "the normalized operation of the parameter w is ignored in Section 5''. Going over Section 5, the only "normalization'' in Section 5 appears in equation (24).
> > Let us elaborate each equality in equation (24), reproduced below, where we compute the cosine of the normalized model $w_{t}$ and the normalized class means $\mu$:
> >
> > $$  \cos\left(\frac{w_t}{\\\|w_t\\\|}, \frac{\mu}{\\\| \mu\\\|} \right)
> > = \frac{  \langle w_t, \mu \rangle   }{ \\\| w_t \\\| \\\| \mu \\\| }
> > = \frac{  \langle w_t, \mu \rangle   }{ b_t  \sqrt{ \\\| \mu \\\|^2 + \langle w_t,\mu \rangle^2 / b_t^2 } }
> > = \mathrm{sign}( r_t )\frac{1}{ \sqrt{1 + \\\| \mu \\\|^2 /r_t^2} }  \quad (24)$$
> >
> > The first equation in (24) is by the definition of the cosine,
> > the second equation in (24) is because
> >
> > $$ \\\| w_t\\\|
> > = \sqrt{ w_t^2[1] + \sum_{i\neq 1} w_t^2[i]  }
> > = \sqrt{ (w_t^\top \mu / \| \mu \|)^2 + b_t^2  },$$
> >
> > where we recall $b_{t}:= \sqrt{\sum_{i\neq 1} w_t^2[i] }$ and
> > $\mu := [ \\\| \mu\\\|, 0, \dots,  0]^{\top}$ (so $w_{t}[1] = w_t^\top \mu / \\\| \mu \\\|$),
> > and hence
> >
> > $$\frac{  \langle w_t, \mu \rangle   }{ \\\| w_t \\\| \\\| \mu \\\| }
> > =  \frac{  \langle w_t, \mu \rangle   }{ \sqrt{ (w_t^\top \mu / \\\| \mu \\\|)^2 + b_t^2  } \\\| \mu \\\| }
> > = \frac{  \langle w_t, \mu \rangle   }{ b_t  \sqrt{ \\\| \mu \\\|^2 + \langle w_t,\mu \rangle^2 / b_t^2 } }. $$
> >
> > The last equation in (24) is by using the definition $r_{t}:= \frac{a_t}{b_t} = \frac{ \langle w_t, \mu \rangle }{ b_t }$ so that
> > $$ \frac{  \langle w_t, \mu \rangle   }{ b_t  \sqrt{ \\\| \mu \\\|^2 + \langle w_t,\mu \rangle^2 / b_t^2 } }
> > = \frac{r_t}{  \sqrt{ \\\| \mu \\\|^2 + r_t^2} }
> > = \mathrm{sign}( r_t )\frac{1}{ \sqrt{1  + \\\| \mu \\\|^2 /r_t^2} }. $$
> >
> > We hope that our response has answered the reviewer's question, but please kindly let us know if we misunderstood the question.
> >
> > __*5. How to derive Equation 24 is not very clear.*__
> >
> > Thanks for raising this concern, which helps improve the presentation of the paper.
> > There was a typo on Equation 24 and we have fixed it in the revised version.
> > Please kindly find our response above.
> >
> > ===
> >
> >
> > It would be appreciated if the reviewer can consider upgrading the score if we have provided satisfying answers. Or, it would also be appreciated if the reviewer can let us know whether we could address any parts in more detail. Thank you.

---

> > > ### Comment · Reviewer_4HLp · 2022-11-24
> > > **Comments on the authors' response**
> > >
> > > Thank the authors for addressing most of my concerns, and they have corrected them in the new submitted version.
> > >
> > > For Q4, I think my question is confusing. In Proposition 1, $a_t:= <w_t, \frac{\mu}{||\mu||}>$, but Section 5 uses a little difference setting that is $a_t:= <w_t, \mu>$. I noticed that Eq.8 defines a model should satisfy the **Condition 1** that contains a normalized operation of $\mu$. So, I am interested in why there is a slight difference in the definition of $a_t$.
> > >
> > > Overall, this work theoretically explains GD with hard and conjugate pseudo-labels for test-time adaptation. Although the assumptions of the paper are a bit too strong, I think the contribution is good for publication.

---

> > > > ### Author Response · Authors · 2022-11-25
> > > > **Thanks for the reply and for clarifying the question Q4**
> > > >
> > > >
> > > > We overloaded the notation $a_{t}$; thanks for pointing this out.
> > > > In the proof of Proposition 1 in Section 4, the quantity that we denote $a_{t}:= \langle w_{t}, \frac{\mu}{\\\| \mu \\\|} \rangle$ is different from the quantity we denote $a_{t}:= \langle w_{t}, \mu \rangle$ in Section 5.
> > > > To avoid this overload, in the next revision we will keep the definition $a_{t} :=\langle w_{t}, \mu \rangle$ as in Section 5, and replace all $a_t$'s in the proof of Proposition 1 by another notation  $\bar{a}_{t} := \frac{a_t}{\\\| \mu \\\|} =
> > > > \frac{ \langle w_t , \mu \rangle }{\\\| \mu \\\|} $.
> > > >
> > > > We note that Condition 1 in the definition of an $\epsilon$-optimal predictor means that $w$ and $\mu$ forms an acute angle, and the angle does not change when we scale down $\mu$ by a factor of $\\\| \mu \\\|$ or scale up $\frac{\mu}{\\\| \mu \\\|}$ by multiplying it with $\\\| \mu \\\|$.
> > > >
> > > >
> > > > Thanks for raising this point, which helps improve the presentation.

---

### Official Review · Reviewer_VqqL · 2022-10-25

**Confidence:** 3
**Correctness:** 4
**Technical Novelty And Significance:** 4
**Empirical Novelty And Significance:** 3
**Recommendation:** 8

**Clarity, Quality, Novelty And Reproducibility:**

The paper is well-motivated and well written. As far as I can tell, the theoretical work is of good quality. I particularly like that the authors are very honest about the scope and the limitations of the work, i.e., the paper actually delivers what is promised. Moreover, I really enjoyed that the main arguments of the proofs (which are arguable the main contribution of the paper) can be found in the main text and are not hidden in the appendix.

**Strength And Weaknesses:**

There has been much work on test-time adaption in recent years. Although certain behaviours are empirically understood, we often do not understand the underlying principles yet.
Although the authors make certain assumptions (e.g., binary labels, population dynamics), these assumptions are probably necessary for coming up with theoretical guarantees and are not too limiting.
The main results are well-explained and technically sound. As such, they do not only provide as with a better understanding, but might spur future research for more general settings.

The empirical evaluation is restricted to a small synthetic example. These experiments corroborate the theoretical findings and as such are sufficient for the scope of this paper. Yet, one could argue that coming up with constructive insights from the theoretical work and showing some improvements on some more realistic test-time adaptation problems would have been nice.

From Algorithm 1 I further take that the samples are all drawn from the same new domain. It can make a drastic difference whether this is the case or if the domain changes more frequently. Although this assumption makes sense in many settings, I miss a brief discussion on it.

**Summary Of The Paper:**

The paper tries to theoretically understand why GD with conjugate labels outperform other pseudo labels. Further results focus on the role of the loss function and provide certain guarantees on the convergence.

**Summary Of The Review:**

The main contribution of the paper is that it provides a novel understanding on the role of the labels (conjugate vs. hard) and the loss function on the potential for test-time adaption. As such, I think it is an interesting theoretical piece of work that might spur further research in this direction.

---

> ### Author Response · Authors · 2022-11-12
> **Thanks for the comments and the positive feedback.**
>
> Thanks for the comments and the positive feedback. Please kindly find our response below.
>
> - __*From Algorithm 1 I further take that the samples are all drawn from the same new domain. It can make a drastic difference whether this is the case or if the domain changes more frequently. Although this assumption makes sense in many settings, I miss a brief discussion on it.*__
>
> Yes, we assume that the unlabeled samples are i.i.d. from the fixed distribution in the new domain. The new domain does not change, which is also the case for the experimental setup of Goyal et al. (2022).
> A scenario where this assumption makes sense is that a model was trained on data collected from a specific institution (e.g., a big private hospital),
> but later it needs to be adapted to another institution (e.g., a veteran hospital).
> We will add more discussions about this assumption in the revision.
> Extending our results to more general domain shifts is a challenging question that we would like to pursue in a future work.
>
> - __*The empirical evaluation is restricted to a small synthetic example. These experiments corroborate the theoretical findings and as such are sufficient for the scope of this paper. Yet, one could argue that coming up with constructive insights from the theoretical work and showing some improvements on some more realistic test-time adaptation problems would have been nice.*__
>
> Thanks for raising this point.
> The work of Goyal et al. (2022) has provided comprehensive experiments evaluating GD with hard labels and conjugate labels on several benchmarks.
> Our work provides a theoretical justification for their results. The analysis and limitations of our results provide clues at what properties of the problem are essential for the phenomena. We will keep exploring whether there are ways to improve the performance of conjugate labels.

---

### Official Review · Reviewer_3i1a · 2022-10-28

**Confidence:** 2
**Correctness:** 3
**Technical Novelty And Significance:** 2
**Empirical Novelty And Significance:** 2
**Recommendation:** 5

**Clarity, Quality, Novelty And Reproducibility:**

**[Clarity and Quality]**
This paper is generally well written.

---
**[Novelty]**
See weakness section.

---
**[Reproducibility]**
Not applicable.


**Strength And Weaknesses:**

**Strength**

**[S1]** This paper is generally well written and easy to follow.

**[S2]** Conjugated pseudo label is timely topic in test-time adaptation. Besides, pseudo-label itself is general techniques and worth investigation.

---
**Weakness**

**[W1] Assumption of the analysis is not tailored for TTA. ** While this paper mainly focus on the test-time adaptation setup (as clearly shown in the title), the current analysis seems to be hold for general pseudo-label setup. In other words, it is unclear what is the uniqueness of the TTA setup, and how the analysis explain the nature of conjugated pseudo-label.

Of course, the generality of the analysis itself is not a bad thing, however, it is unclear for me how the analysis is significant if it is not tailored for TTA. The paper should clearly discuss whether the current analysis is specialized for TTA setup, or more general one. Besides, if the later case, the paper should discuss more on what was already well investigated in pseudo-labeling and how the analysis in this paper is significant compared to existing findings.


**[W2]Practical implication is limited. ** Since the analysis conducted only for the simple case on binary classification with Gaussian model, practical implication is limited. They also provide only limited toy empirical investigations.

**Summary Of The Paper:**

**Summary**
This paper investigates why conjugated pseudo label in test-time adaptation (TTA), which is recently proposed approach, perform better than hard pseudo-labeling. Specifically, they show that under Gaussian model, GD with hard pseudo-labels fails to find optimal solution while GD with conjugated pseudo label reaches optimal solution.


**Summary Of The Review:**

Overall, I agree that the conjugated pseudo label is timely and worth to investigate topic, but the current analysis is limited to very limited setup and therefore the practical impact is limited. Since I am not the theory person, I might underestimate the significance of the theoretical advancement.

---

> ### Author Response · Authors · 2022-11-12
> **Thanks for the comments and feedback (1/2)**
>
> Thanks for the comments and feedback.
>
> __*W1: Assumption of the analysis is not tailored for TTA. While this paper mainly focus on the test-time adaptation setup (as clearly shown in the title), the current analysis seems to be hold for general pseudo-label setup. In other words, it is unclear what is the uniqueness of the TTA setup, and how the analysis explain the nature of conjugated pseudo-label.
> Of course, the generality of the analysis itself is not a bad thing, however, it is unclear for me how the analysis is significant if it is not tailored for TTA. The paper should clearly discuss whether the current analysis is specialized for TTA setup, or more general one. Besides, if the later case, the paper should discuss more on what was already well investigated in pseudo-labeling and how the analysis in this paper is significant compared to existing findings.*__
>
> Thanks for the comment. First of all, we would like to be clear that the setup is about analyzing GD with hard and conjugate labels for fully test-time adaptation
> on the binary classification problem under Gaussian model.
> __In other words, our setup is about a specific methodology (i.e., GD with pseudo-labels) for fully test-time adaptation.__
> As mentioned in the beginning of the introduction of this paper, the problem of fully test-time adaptation is about adapting a model from a source domain so that it fits to a new domain at test time, without access to the true labels of samples from the new domain
> nor the data from the source domain.
> The problem was studied in the work of Goyal et al. (2022), who propose conjugate pseudo-labels, and was also investigated in a prior theory work of
> Chen et al. (2020).
> Goyal et al. (2022) consider applying GD with pseudo-labels to adapting the source model to the new domain for tackling this problem. In Appendix A of Goyal et al. (2022), they report an interesting experimental result for test-time adaptation under Gaussian model, which shows that GD with conjugates-labels outperforms GD with hard-labels for exponential loss empirically, albeit without theoretical analysis. This gap motivates us to analyze GD for exponential loss and other losses like square loss and logistic loss with hard and conjugate labels for the binary classification problem.
> Second, as discussed in Section 4, we assume that the source model forms an acute angle with the class mean $\mu$ in the target domain, which implies that learning from the source distribution helps with learning from the new distribution in the target domain. This is where the relation between the source domain and the new domain is exploited. It is noted that a similar setup appears in a prior theory  work of
> Chen et al. (2020), as we mentioned in the introduction of this paper.
> Specifically, Chen et al. (2020) provably show that when data have spurious features
> under a variant of Gaussian model, if projected GD is initialized from a source model which has sufficiently high accuracy in a new domain, then by minimizing the exponential loss with hard pseudo-labels, projected GD would successfully converge to an approximately optimal solution in the new domain under certain conditions.
> Another theoretical work of domain adaptation by Kumar et al. (2020) also made some relevant assumptions under Gaussian model so that the classifier learned from the previous distribution is not too different from the best one for the current distribution, which was also described and discussed in detail in Section 3 of this paper.
>
> Last but not least, we believe that there is little theoretical analysis of GD with pseudo-labels in the literature. The only one that we know is the work of Chen et al. (2020) in the context of test-time adaptation, and we have discussed the differences in the paper.
>
> __References:__
>
> [1] Sachin Goyal, Mingjie Sun, Aditi Raghunathan, and Zico Kolter. Test-Time Adaptation via Conjugate Pseudo-labels. NeurIPS, 2022.
>
> [2] Yining Chen, Colin Wei, Ananya Kumar, and Tengyu Ma. Self-training avoids using spurious features under domain shift. NeurIPS, 2020.
>
> [3] Ananya Kumar, Tengyu Ma, and Percy Liang. Understanding Self-Training for Gradual Domain Adaptation. ICML, 2020.

---

> > ### Comment · Reviewer_3i1a · 2022-12-02
> > **Reply**
> >
> > Sorry for the late response. Thank you for your rebuttal. I have read the rebuttal and other reviews. It made me understand the theoretical insights more clearly, so I increased my score from 3 to 5.
> >
> > I still have concerns about the practical impacts of the paper, which is why I hesitate to support the paper strongly. As mentioned in the original review, the empirical evaluations are very limited. There is no evaluation of standard datasets (e.g., ImageNet-C, Cifar10-C, etc.) used in other papers. Besides, the simple pseudo-labels, which are analyzed in the paper, are not used in practice. It is usually accompanied by some heuristics, such as confidence thresholding (e.g.,  the original TENT and conjugated pseudo-label form use confidence thresholding for baselines) or some sort of ensembling (e.g. [1]). I'm curious whether using such techniques would affect the theoretical analysis and some empirical analyses.
> >
> > Finally, I recommend the authors to better define the problem setup in detail. As authors might be noticed, there are lots of slight differences, e.g., online/offline and stable/dynamic distribution changes, even if you just say "test-time-adaptation". I'm also curious whether these detailed differences affect the conclusion of the paper or not.
> >
> > Overall, I still have concerns about the practical impacts, which is why I rated the paper as 5. However, I don't have a strong objection to accepting the paper as well.
> >
> > [1] Continual Test-Time Domain Adaptation, https://arxiv.org/abs/2203.13591

---

> > > ### Author Response · Authors · 2022-12-04
> > > **Thanks for the feedback and upgrading the score. (1/2)**
> > >
> > > Thanks for the feedback and upgrading the score. Please kindly find our response to each of the comments below.
> > >
> > > __*I still have concerns about the practical impacts of the paper, which is why I hesitate to support the paper strongly. As mentioned in the original review, the empirical evaluations are very limited. There is no evaluation of standard datasets (e.g., ImageNet-C, Cifar10-C, etc.) used in other papers.*__
> > >
> > > We would like to clarify again that the work of Goyal et al. (2022) has provided comprehensive experiments in comparing GD with hard labels and conjugate labels on many domain adaptation datasets including ImageNet-C and Cifar10-C. Our work is concerned with the theoretical aspect, as the theoretical analysis of GD with pseudo-labels (not necessarily conjugate labels) for test-time adaptation is sparse in the literature.
> > > Our theoretical results complement and champion the empirical results of Goyal et al. (2022) and might spur future research in this direction, as commented by other reviewers.
> > >
> > >
> > > __*Besides, the simple pseudo-labels, which are analyzed in the paper, are not used in practice. It is usually accompanied by some heuristics, such as confidence thresholding (e.g., the original TENT and conjugated pseudo-label form use confidence thresholding for baselines) or some sort of ensembling (e.g. [1]).
> > > I'm curious whether using such techniques would affect the theoretical analysis and some empirical analyses.*__
> > >
> > > Respectfully, we believe the claim that ``the simple pseudo-labels, which are analyzed in the paper, are not used in practice'' is not quite right. The work of Goyal et al. (2022) shows the promising empirical performance of GD with conjugate-label without coupling any other heuristics like confidence thresholding or ensembling.
> > > On the other hand, the baseline GD with hard labels in their experiments was combined with confidence thresholding (i.e., not updating when the predicted score is below a threshold) as the reviewer kindly points out.
> > >
> > > While studying the effect of any heuristics on top of GD + pseudo-labels is not the focus of this work,
> > > we would like to provide a sketch on how
> > > our theoretical analysis can be modified to accommodate GD + pseudo-labels + confidence thresholding as follows.
> > > Denote the threshold of not updating the model as $\tau$.
> > > Then, the probability of not updating the model $w_{t}$ is
> > >
> > > $$ \Pr\left( -\tau \leq  \langle w_t, x \rangle  \leq \tau  \right)  = \frac{1}{2}
> > >  \Pr \left( -\frac{\tau}{\\\| w_t\\\|} \leq N\left( \frac{ \mu^\top w_t}{ \sigma \\\|  w_t \\\| }  , 1\right) \leq  \frac{\tau}{\\\| w_t \\\|}  \right) + \frac{1}{2}  \Pr \left( -\frac{\tau}{\\\| w_t\\\|} \leq N\left( \frac{ - \mu^\top w_t}{ \sigma \\\|  w_t \\\| }  , 1\right) \leq  \frac{\tau}{\\\| w_t \\\|}  \right):= \Gamma \left(  w_t; \sigma, \mu \right).$$
> > >
> > > With this, one can proceed to analyze the resulting dynamic, i.e.,
> > > with probability $1-\Gamma \left(  w_t; \sigma, \mu \right)$,
> > > the dynamic is as the ones in our analysis, while
> > > with probability $\Gamma \left(  w_t; \sigma, \mu \right)$, it does not update and therefore the quantities of the interest do not change.
> > >
> > > We would like to note that while theoretically understanding when and why confidence thresholding (or ensembling) works is not the subject of this paper, we agree that this is a promising direction. As we also mentioned in the conclusion, theoretically understanding GD with other pseudo-labels or other domain adaptation techniques could be promising. Even analyzing confidence thresholding alone might be a valuable contribution in learning theory, as there is little work in the literature as far as we are aware.
> > >
> > > __*Finally, I recommend the authors to better define the problem setup in detail. As authors might be noticed, there are lots of slight differences, e.g., online/offline and stable/dynamic distribution changes, even if you just say "test-time-adaptation". I'm also curious whether these detailed differences affect the conclusion of the paper or not.*__
> > >
> > > Thanks for the suggestion. We will be happy to describe the problem setup and other scenarios that the authors mentioned in more detail.
> > >
> > > We would like to clarify that the first paragraph in this paper has outlined our problem setup. Moreover, Section 3 describes the specific  framework/setup of fully test-time adaptation in detail.
> > > We also discuss on Page 4 about the work of Kumar et al. (2020) and highlight that their work considers a scenario of gradual distribution shifts (with the assumption of access to minimizers of some population risks), which is different from the setup in our work. Hence, we believe that we have clearly described the setting of fully test time adaptation in this paper. Nevertheless, we will improve the presentation and provide a more thorough literature review including the heuristics that the reviewer mentions in the next version.
> > >
> > > (to continue)

---

> > > > ### Author Response · Authors · 2022-12-04
> > > > **Thanks for the feedback and upgrading the score. (2/2)**
> > > >
> > > > (following the last comment)
> > > >
> > > > Under the setting in Section 3, we analyze GD with pseudo-labels with the most common loss functions for the task of binary classification, i.e., square loss, logistic loss, and exponential loss. Our conclusions and theoretical results are about minimizing the resulting self-training loss functions via GD under the setting accordingly, and hence they do not cover other scenarios like ever-changing data distributions. We believe that analyzing GD with pseudo labels in other scenarios can be an interesting future direction of research that builds on our work.
> > > >
> > > > ===
> > > >
> > > >
> > > > We will appreciate it if the reviewer can let us know if we have clarified the concerns, and/or if the reviewer is more positive/supportive about this work. Thank you.

---

> ### Author Response · Authors · 2022-11-12
> **Thanks for the comments and feedback (2/2)**
>
> Thanks for the comments and feedback.
>
> __*W2: Practical implication is limited. Since the analysis conducted only for the simple case on binary classification with Gaussian model, practical implication is limited. They also provide only limited toy empirical investigations.*__
>
> We would like to clarify that the work of Goyal et al. (2021) has provided comprehensive experiments in comparing GD with hard labels and conjugate labels on many domain adaptation datasets.
> Our paper attempts to provide a theory for the observed results, by analyzing GD
> with hard labels and conjugate labels
> for the common loss functions (i.e., square loss, logistic loss, and exp loss) under Gaussian model.
> We prove that GD with conjugate-labels converges exponentially fast to an optimal predictor in the new domain under Gaussian model (Section 5.1), while GD with hard-labels fails in this task (Section 4). We also analyze GD with hard-labels and GD with conjugate labels for logistic loss and exponential loss, and we derive their convergence rates (Section 5.2).
> We further discuss how our theoretical result in Section 5.2 predicts that the use of conjugate labels can lead to a faster convergence of GD than that of the use of hard labels, as our theoretical result reveals a connection between the tail of the resulting self-training loss function and the convergence rate of GD.
> Our work is the first to systematically analyze GD with hard-labels and conjugate labels for test-time adaptation, and is the first to provably show the advantage of GD with conjugate labels (and hence champions the work of Goyal et al. (2022)),
> though under a simple theoretical framework of Gaussian model. However, as we discussed in the paper, there have been some theory works in machine learning literature that adopt the framework of simple Gaussian model or its variants for  analysis, which include the aforementioned works of Chen et al. (2020) and Kumar et al. (2020) as well as Schmidt et al. (2018) and Carmon et al. (2019) for adversarial robustness.
> A recent work of Nguyen et al. (2022) also adopts a simple variant of Gaussian model for analyzing large-scale web-based datasets for pre-training.
>
> We believe that analyzing complicated phenomena under a simple model/framework is quite common in theoretical works, and we view it as a trait rather than a weakness of our work.
> The other reviewers seem to appreciate our work and give our work positive feedback.
> Reviewer VqqL writes
> "
> The main contribution of the paper is that it provides a novel understanding on the role of the labels (conjugate vs. hard) and the loss function on the potential for test-time adaption. As such, I think it is an interesting theoretical piece of work that might spur further research in this direction.''.
> Reviewer 4HLP writes
> "This paper gives an interesting understanding of the GD with hard and conjugate labels for test-time training. I think this will bring new insights to the community.''.
> Reviewer wLJR writes
> "The paper provides insights into when and why GD with hard labels or conjugate labels works in test-time adaptation.''
>
> Our work is in no way a perfect one. But, as we mentioned, there is little theory result in this direction, and bridging the gap between theory and practice needs to start somewhere. We are excited that the current work makes a solid step forward in this direction. It would be appreciated if the reviewer can consider upgrading the score if our response is reasonable. Thank you.
>
> __References:__
>
> [1] Yining Chen, Colin Wei, Ananya Kumar, and Tengyu Ma. Self-training avoids using spurious features under domain shift. NeurIPS, 2020.
>
> [2] Ananya Kumar, Tengyu Ma, and Percy Liang. Understanding Self-Training for Gradual Domain Adaptation. ICML, 2020.
>
> [3] Ludwig Schmidt, Shibani Santurkar, Dimitris Tsipras, Kunal Talwar, and Aleksander Madry. Adversarially
> Robust Generalization Requires More Data. NeurIPS, 2018.
>
> [4] Yair Carmon, Aditi Raghunathan, Ludwig Schmidt, John C Duchi, and Percy S Liang. Unlabeled data improves adversarial robustness. NeurIPS, 2019
>
> [5] Thao Nguyen, Gabriel Ilharco, Mitchell Wortsman, Sewoong Oh, Ludwig Schmidt.
> Quality Not Quantity: On the Interaction between Dataset Design and Robustness of CLIP. NeurIPS 2022

---

> ### Author Response · Authors · 2022-12-02
> **Dear Reviewer 3i1a**
>
> We would pretty much appreciate it if the reviewer can engage with us, and kindly let us know if we have clarified all the concerns and provided satisfying answers. Or, let us know if we should address any parts in more detail, and we will be happy to do so. Thank you.
>
> All the best,
>
> the authors

---

### Decision · Program_Chairs · 2023-01-20

**Decision:**

Accept: poster

**Justification For Why Not Higher Score:**

The reviewers were overall positive, but one reviewer stayed on the fence and there remain some concerns about assumptions made in the paper (along with practicality of the results).

**Justification For Why Not Lower Score:**

In the end, three of the four reviewers advocated for accepting the paper, and the weaknesses noted by the reviewers seem not large enough to warrant rejecting the paper.

**Metareview: Summary, Strengths And Weaknesses:**

Thanks for your submission to ICLR.

Three of the four reviewers were positive about the paper, and the fourth reviewer was borderline by the end of the discussion period.  Strengths of the paper: the paper is well-written, on an interesting topic, and contains an interesting theoretical analysis.  Weaknesses: several reviewers noted some issues with assumptions required for the theory, limited practicality, and a somewhat restrictive empirical evaluation.

The most negative reviewer did raise their score during discussion, and indicated that they were not opposed to accepting the paper.  The authors did a good job in responding to the concerns of the reviewers.  Thus, I am recommending accepting the paper.

**Note From Pc:**

if the above contains the word "oral" or "spotlight" please see: "oral" presentation means -> notable-top-5% and "spotlight" means -> notable-top-25%. As stated in our emails, we are disassociating presentation type from AC recommendations